# HD-Painter: High-Resolution and Prompt-Faithful Text-Guided Image Inpainting with Diffusion Models

**Hayk Manukyan**[1][*]    **Andranik Sargsyan**[1][*]    **Barsegh Atanyan**[1]    **Zhangyang Wang**[2]
**Shant Navasardyan**[1]    **Humphrey Shi**[1,3]

[1]Picsart AI Research (PAIR)    [2]UT Austin    [3]Georgia Tech

## Abstract

Recent progress in text-guided image inpainting, based on the unprecedented success of text-to-image diffusion models, has led to exceptionally realistic and visually plausible results. However, there is still significant potential for improvement in current text-to-image inpainting models, particularly in better aligning the inpainted area with user prompts. Therefore, we introduce *HD-Painter*, a **training-free** approach that **accurately follows prompts**. To this end, we design the *Prompt-Aware Introverted Attention (PAIntA)* layer enhancing self-attention scores by prompt information resulting in better text aligned generations. To further improve the prompt coherence we introduce the *Reweighting Attention Score Guidance (RASG)* mechanism seamlessly integrating a post-hoc sampling strategy into the general form of DDIM to prevent out-of-distribution latent shifts. Our experiments demonstrate that HD-Painter surpasses existing state-of-the-art approaches quantitatively and qualitatively across multiple metrics and a user study. Code is publicly available at: **https://github.com/Picsart-AI-Research/HD-Painter**.

## 1 Introduction

The recent wave of diffusion models (Ho et al., 2020; Song et al., 2021) has taken the world by storm, becoming an increasingly integral part of our everyday lives. After the unprecedented success of text-to-image models (Rombach et al., 2022; Ramesh et al., 2022; Saharia et al., 2022; Wu et al., 2022) diffusion-based image manipulations such as prompt-conditioned editing (Hertz et al., 2022; Brooks et al., 2023), controllable generation (Zhang & Agrawala, 2023; Mou et al., 2023), personalized and specialized image synthesis (Ruiz et al., 2023; Gal et al., 2022; Lu et al., 2023) became hot topics in computer vision leading to a huge amount of applications. Particularly, text-guided image completion or inpainting (Wang et al., 2023; Wu et al., 2022; Avrahami et al., 2022) allows users to generate new content in user-specified regions of given images based on textual prompts, leading to use cases like retouching specific areas of an image, replacing or adding objects, and modifying subject attributes such as clothes, colors, or emotion.

Pretrained text-to-image generation models such as Stable Diffusion (Rombach et al., 2022), Imagen (Saharia et al., 2022), and Dall-E 2 (Ramesh et al., 2022) can be adapted for image completion by blending diffused known regions with generated (denoised) unknown regions during the backward diffusion process. Although such approaches (Avrahami et al., 2022; 2023) produce visually plausible completions, they are not well harmonized and lack global scene understanding, especially when denoising in high diffusion timesteps.

To address this, existing methods (Rombach et al., 2022; Nichol et al., 2021; Podell et al., 2023; Saharia et al., 2022), modify pretrained text-to-image models to take additional context information and fine-tune specifically for text-guided image completion. GLIDE (Nichol et al., 2021) and Stable Inpainting (Rombach et al., 2022) concatenate the mask and the masked image as additional channels to the input of the diffusion UNet, initializing the new convolutional weights with zeros, then fine tune the modified model using random masks together with the initial prompt.

---

[*] Indicates equal contribution.

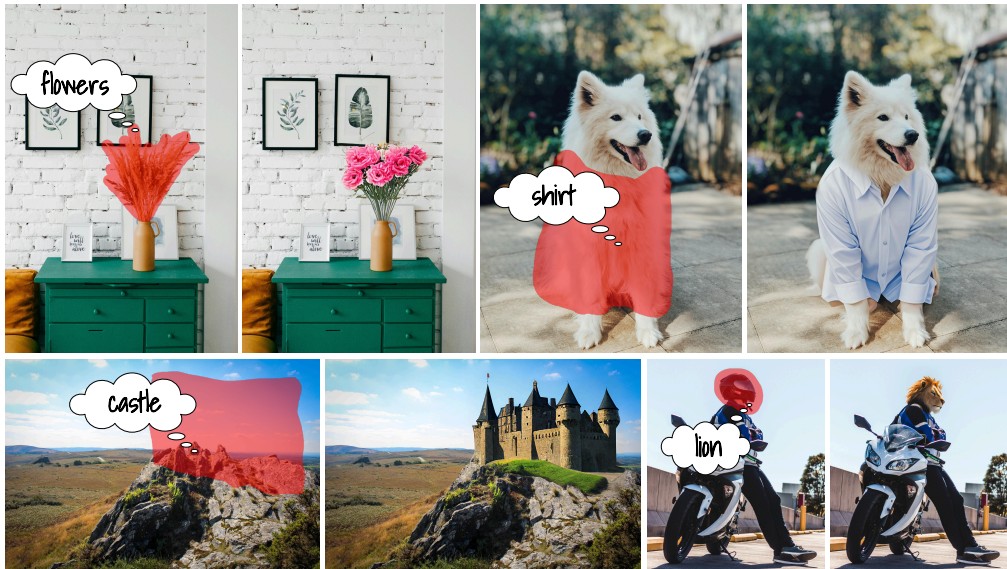

Figure 1: Inpainting results with our approach. Results have been upscaled to 2048px large side using our developed inpainting-specialized super-resolution technique. The method is able to faithfully fill the masked region according to the prompt even if the combination of the prompt and the known region is highly unlikely. Zoom in to view details.

However, SmartBrush (Xie et al., 2023) and Imagen Editor (Wang et al., 2023) mention the weak image-text alignment of such models, attributing it to the random masking strategies, and the misalignment of the global prompts used during training with the local context of the masked region. In this paper, we will address this issue as *prompt neglect*. To alleviate this problem, both papers introduce novel, object-aware masking strategies. Additionally SmartBrush proposes BLIP captioning approach, to ensure a better alignment of the inpainting prompt with the masked region. Nonetheless, we find that while this approach reduces prompt neglect, it also decreases the generation quality.

We notice that prompt neglect is commonly expressed in two ways: either the model fills in the masked region with background (*background dominance*, fig. 5, columns 1, 3, 5), or the model completes a nearby object partially occluded by the mask (*nearby object dominance*, fig. 5, columns 2, 4, 6). In both cases the issue seems to be caused by the model preferring the local context of the known region to the textual information provided by the prompt.

To address the mentioned problems we introduce *Prompt-Aware Introverted Attention (PAIntA)*, without any *training or fine-tuning* requirements. PAIntA enhances the self-attention scores according to the given textual condition aiming to decrease the impact of non-prompt-relevant information from the image known region while increasing the contribution of the prompt-aligned known pixels.

To improve the text-alignment of the generation results even further we apply a *post-hoc guidance* mechanism by leveraging the cross-attention scores. However the vanilla post-hoc guidance mechanism used by seminal works such as (Dhariwal & Nichol, 2021; Epstein et al., 2023) may lead to generation quality degradation due to out-of-distribution shifts caused by the additional gradient term in the backward diffusion equation. To this end we propose *Reweighting Attention Score Guidance (RASG)*, a post-hoc mechanism seamlessly integrating the gradient component in the general form of DDIM process. This allows to simultaneously guide the sampling towards more prompt-aligned latents and keep them in their trained domain leading to visually plausible inpainting results.

To summarize, our main contributions are as follows:

- We introduce the *Prompt-Aware Introverted Attention (PAIntA)* layer to alleviate the prompt neglect issues of background and nearby object dominance in text-guided image inpainting.

- To further improve the text-alignment of generation we present the *Reweighting Attention Score Guidance (RASG)* strategy which enables to prevent out-of-distribution shifts while performing post-hoc guided sampling.

- Our pipeline for text-guided image completion is *training-free* and demonstrates a significant advantage over current state-of-the-art approaches quantitatively and qualitatively.

## 2 RELATED WORK

### 2.1 IMAGE INPAINTING

Early deep learning approaches for image inpainting (Yu et al., 2018; Yi et al., 2020; Navasardyan & Ohanyan, 2020) introduce mechanisms to propagate deep features from known regions. Later (Zhao et al., 2021; Zheng et al., 2022; Xu et al., 2023; Sargsyan et al., 2023) utilize StyleGAN-v2-like (Karras et al., 2020) decoder and discriminative training for better image detail generation.

Image inpainting also benefited from diffusion models, particularly with the emergence of text-guided inpainting. Given a pre-trained text-to-image diffusion model Avrahami et al. (2022; 2023) replace the unmasked region of the latent by the noised version of the known region during sampling. Nichol et al. (2021); Wang et al. (2023); Podell et al. (2023); Xie et al. (2023) fine-tune pre-trained text-to-image models for text-guided image inpainting by conditioning the denoising model on the inpainting mask and the known region, concatenating them with the input latents. Zhang & Agrawala (2023) obtain an inpainting model by attaching trainable modules to the UNet, while keeping the base model unchanged. We propose a training-free approach leveraging plug-and-play components PAIntA and RASG, improving text-prompt alignment.

### 2.2 INPAINTING-SPECIFIC ARCHITECTURAL BLOCKS

Early deep learning approaches were designing special layers for better/more efficient inpainting. Particularly, Liu et al. (2018); Yu et al. (2019); Navasardyan & Ohanyan (2020) introduce special convolutional layers dealing with the known region of the image to effectively extract the information useful for visually plausible image completion. Yi et al. (2020) introduce the contextual attention layer reducing the unnecessarily heavy computations of all-to-all self-attention for high-quality inpainting. In this work we propose Prompt-Aware Introverted Attention (PAIntA) layer, specifically designed for text-guided image inpainting. It aims to decrease (increase) the prompt-irrelevant (-relevant) information from the known region for better text aligned inpainting generation.

### 2.3 POST-HOC GUIDANCE IN DIFFUSION PROCESS

Post-hoc guidance methods are backward diffusion sampling techniques which guide the next step latent prediction towards a specific objective function minimization. Particularly Dhariwal & Nichol (2021) introduced classifier-guidance aiming to generate images of a specific class. Later CLIP-guidance was introduced by Nichol et al. (2021) leveraging CLIP (Radford et al., 2021) as an open-vocabulary classification method. Chefer et al. (2023) guide image generation by maximizing the maximal cross-attention score relying on multi-iterative optimization process resulting in more text aligned results. Epstein et al. (2023) utilizes the cross-attention scores for object position, size, shape, and appearance guidances. All the mentioned post-hoc guidance methods shift the latent generation process by a gradient term (see eq. (7)) sometimes leading to image quality degradations.

To this end we propose the Reweighting Attention Score Guidance (RASG) mechanism allowing to perform post-hoc guidance with any objective function **while preserving the diffusion latent domain**. Specifically for inpainting, to alleviate the issue of prompt neglect, we benefit from a guidance objective function based on the open-vocabulary segmentation properties of cross-attentions.

## 3 METHOD

We first formulate the text-guided image completion problem followed by an introduction to diffusion models, particularly Stable Diffusion and Stable Inpainting (Rombach et al., 2022). We then discuss the overview of our method and its components. Afterwards we present our Prompt-Aware Introverted Attention (PAIntA) block and Reweighting Attention Score Guidance (RASG) mechanism in detail.

Let $I \in \mathbb{R}^{H \times W \times 3}$ be an RGB image, $M \in \{0, 1\}^{H \times W}$ be a binary mask indicating the region in $I$ one wants to inpaint with a textual prompt $\tau$. The goal of text-guided image inpainting is to output an image $I^c \in \mathbb{R}^{H \times W \times 3}$ such that $I^c$ contains the objects described by the prompt $\tau$ in the region $M$ while outside $M$ it coincides with $I$, i.e. $I^c \odot (1 - M) = I \odot (1 - M)$.

## 3.1 STABLE DIFFUSION AND STABLE INPAINTING

Stable Diffusion (SD) is a diffusion model that functions within the latent space of an autoencoder $\mathcal{D}(\mathcal{E}(\cdot))$ (VQ-GAN (Esser et al., 2021) or VQ-VAE (Van Den Oord et al., 2017)) where $\mathcal{E}$ denotes the encoder and $\mathcal{D}$ the corresponding decoder. Specifically, let $I \in \mathbb{R}^{H \times W \times 3}$ be an image and $x_0 = \mathcal{E}(I)$, consider the following forward diffusion process with hyperparameters $\{\beta_t\}_{t=1}^T \subset [0, 1]$:

$$q(x_t|x_{t-1}) = \mathcal{N}(x_t; \sqrt{1 - \beta_t} x_{t-1}, \beta_t I), \ t = 1, .., T \tag{1}$$

where $q(x_t|x_{t-1})$ is the conditional density of $x_t$ given $x_{t-1}$, and $\{x_t\}_{t=0}^T$ is a Markov chain. Here $T$ is large enough to allow an assumption $x_T \sim \mathcal{N}(\mathbf{0}, \mathbf{1})$. Then SD learns a backward process (below similarly, $\{x_t\}_{t=T}^0$ is a Markov chain)

$$p_\theta(x_{t-1}|x_t) = \mathcal{N}(x_{t-1}; \mu_\theta(x_t, t), \sigma_t \mathbf{1}), \ t = T, .., 1, \tag{2}$$

and hyperparameters $\{\sigma_t\}_{t=1}^T$, allowing the generation of a signal $x_0$ from the standard Gaussian noise $x_T$. Here $\mu_\theta(x_t, t)$ is defined by the predicted noise $\epsilon_\theta^t(x_t)$ modeled as a neural network (Ho et al., 2020): $\mu_\theta(x_t, t) = \frac{1}{\sqrt{\beta_t}} \left( x_t - \frac{\beta_t}{\sqrt{1 - \alpha_t}} \epsilon_\theta^t(x_t) \right)$. Then $\hat{I} = \mathcal{D}(x_0)$ is returned.

The following claim can be derived from the main DDIM principle, (Song et al., 2021), Theorem 1.

CLAIM 1 *After training the diffusion backward process (eq. (2)) the following $\{\sigma_t\}_{t=1}^T$-parameterized family of DDIM sampling processes can be applied to generate high-quality images:*

$$x_{t-1} = \sqrt{\alpha_{t-1}} \frac{x_t - \sqrt{1 - \alpha_t} \epsilon_\theta^t(x_t)}{\sqrt{\alpha_t}} + \sqrt{1 - \alpha_{t-1} - \sigma_t^2} \epsilon_\theta^t(x_t) + \sigma_t \epsilon_t, \tag{3}$$

*where $\epsilon_t \sim \mathcal{N}(\mathbf{0}, \mathbf{1})$, $\alpha_t = \prod_{i=1}^t (1 - \beta_i)$, and $0 \leq \sigma_t \leq \sqrt{1 - \alpha_{t-1}}$ can be arbitrary parameters.*

Usually (e.g. in SD or Stable Inpainting described below) $\sigma_t = 0$ is taken to get a deterministic process:

$$x_{t-1} = \sqrt{\alpha_{t-1}} \left( \frac{x_t - \sqrt{1 - \alpha_t} \epsilon_\theta^t(x_t)}{\sqrt{\alpha_t}} \right) + \sqrt{1 - \alpha_{t-1}} \epsilon_\theta^t(x_t), \ t = T, \ldots, 1. \tag{4}$$

For text-to-image synthesis, SD guides the processes with a textual prompt $\tau$. Hence the function $\epsilon_\theta^t(x_t) = \epsilon_\theta^t(x_t, \tau)$, modeled by a UNet-like (Ronneberger et al., 2015) architecture, is also conditioned on $\tau$ by its cross-attention layers. For simplicity sometimes we skip $\tau$ in writing $\epsilon_\theta^t(x_t, \tau)$.

As mentioned earlier, Stable DIffusion can be modified and fine-tuned for text-guided image inpainting. To do so Rombach et al. (2022) concatenate the features of the masked image $I^M = I \odot (1 - M)$ obtained by the encoder $\mathcal{E}$, and the (downscaled) binary mask $M$ to the latents $x_t$ and feed the resulting tensor to the UNet to get the estimated noise $\epsilon_\theta^t([x_t, \mathcal{E}(I^M), down(M)], \tau)$, where $down$ is the downscaling operation to match the shape of the latent $x_t$. Newly added convolutional filters are initialized with zeros while the rest of the UNet from a pretrained checkpoint of Stable Diffusion. Training is done by randomly masking images and optimizing the model to reconstruct them based on image captions from the LAION-5B (Schuhmann et al., 2022) dataset. The resulting model shows visually plausible image completion and we refer to it as *Stable Inpainting*.

## 3.2 HD-PAINTER: OVERVIEW

The overview of HD-Painter is presented in fig. 2. To complete the missing region $M$ according to the prompt $\tau$ we take a pre-trained inpainting diffusion model, replace the self-attention layers by PAIntA layers, and perform a diffusion backward process with our RASG mechanism. After getting the estimated latent $x_0$, it is decoded resulting in an inpainted image $I^c = \mathcal{D}(x_0) \in \mathbb{R}^{H \times W \times 3}$.

Additionally, by leveraging high-resolution diffusion models and time-iterative blending, we design a simple yet effective pipeline for up to $2048 \times 2048$px resolution inpainting (see Appendix E).

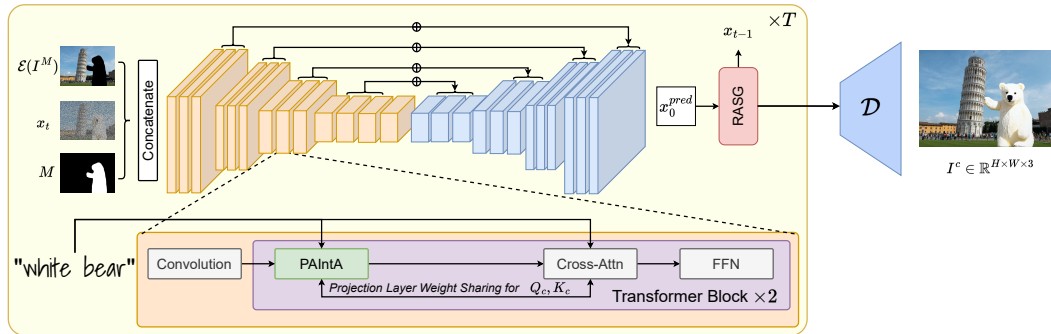

Figure 2: In each diffusion step we denoise the latent $x_t$ by conditioning on the inpainting mask $M$ and the masked downscaled image $I^M = down(I) \odot (1 - M) \in \mathbb{R}^{H \times W \times 3}$ (encoded with the VAE encoder $\mathcal{E}$). To make better alignment with the given prompt our *PAIntA* block is applied instead of self-attention layers. After predicting the denoised $x_0^{pred}$ in each step $t$, we provide it to our *RASG* guidance mechanism to estimate the next latent $x_{t-1}$.

### 3.3 PROMPT-AWARE INTROVERTED ATTENTION (PAINTA)

A thorough analysis of existing text-guided image inpainting methods shows that they often overlook user-provided prompts, relying instead on the surrounding visual context of the inpainting area. For example, in fig. 5, most existing methods fail to create a vase matching the prompt or to generate the boat. We hypothesize that the *visual context dominance* over the prompt is attributed to the *prompt-free, only-spatial* nature of self-attention layers. To support this we visualize the self-attention scores in fig. 3 (a). The heatmaps illustrate how, on average, pixels in the masked region attend to other pixels in the image (see Appendix B for implementation details). For example in the forth row (the image with the vase) we see that the masked region pixels have high attention scores not only with themselves but also with other background pixels. This results in generated pixels that closely resemble the background, undermining the intended generation (given by the prompt). This behavior highlights a limitation of the self-attention layer: it not only disregards the prompt context (as it does not incorporate the prompt as input) but also reinforces similarity between the generated region and the background. Therefore, to alleviate the issue, we introduce plug-in replacement for self-attention, Prompt-Aware Introverted Attention (PAIntA, see fig. 4 (a)) which adjusts the attention scores between mask pixels $i$ and non-mask pixels $j$ according to the alignment of the pixels $j$ with the prompt (higher when more aligned, lower otherwise). fig. 3 (b) shows the attention maps of PAIntA and here we clearly see that the mask region mostly attends to itself by so allowing the prompt to influence the generation process later. This leads to outputs where the desired objects are accurately generated. Below we discuss PAIntA in detail.

Let $X \in \mathbb{R}^{(h \times w) \times d}$ be the input tensor of PAIntA. Similar to self-attention, PAIntA first applies projection layers to get the queries, keys, and values we denote by $Q_s, K_s, V_s \in \mathbb{R}^{(h \times w) \times d}$ respectively, and the similarity matrix $A_{self} = \frac{Q_s K_s^T}{\sqrt{d}} \in \mathbb{R}^{hw \times hw}$. Then, as discussed above, we mitigate the too strong influence of the known region over the unknown by adjusting the corresponding attention scores. To do so, for each unknown pixel $i$ and known region pixel $j$ we multiply the attention scores $(A_{self})_{i,j}$ by a factor $c_j \in [0, 1]$. $c_j$ represents the amount of how much we want to suppress the impact of the known region pixel $j$ on the completion of the missing region. As we want the generation in the missing region to be more aligned with the provided textual prompt, we set $c_j$ based on the similarity between $j$ and the prompt in the embedding space. In other words, we set $c_j$ low for such pixels $j$ from the known region that are not semantically close to the given prompt, and we set $c_j$ high otherwise. Specifically, leveraging the prompt $\tau$, PAIntA defines a new similarity matrix:

$$\tilde{A}_{self} \in \mathbb{R}^{hw \times hw}, \quad (\tilde{A}_{self})_{ij} = \begin{cases} c_j \cdot (A_{self})_{ij} & M_i = 1 \text{ and } M_j = 0, \\ (A_{self})_{ij} & \text{otherwise}, \end{cases} \quad (5)$$

where $M$ is the resized and flattened input mask , and $c_j$ is defined as follows.

We define $\{c_j\}_{j=1}^{hw}$ using the cross-attention spatio-textual similarity matrix $S_{cross} = SoftMax(Q_c K_c^T / \sqrt{d})$, where $Q_c \in \mathbb{R}^{(h \times w) \times d}$, $K_c \in \mathbb{R}^{l \times d}$ are query and key tensors of cor-

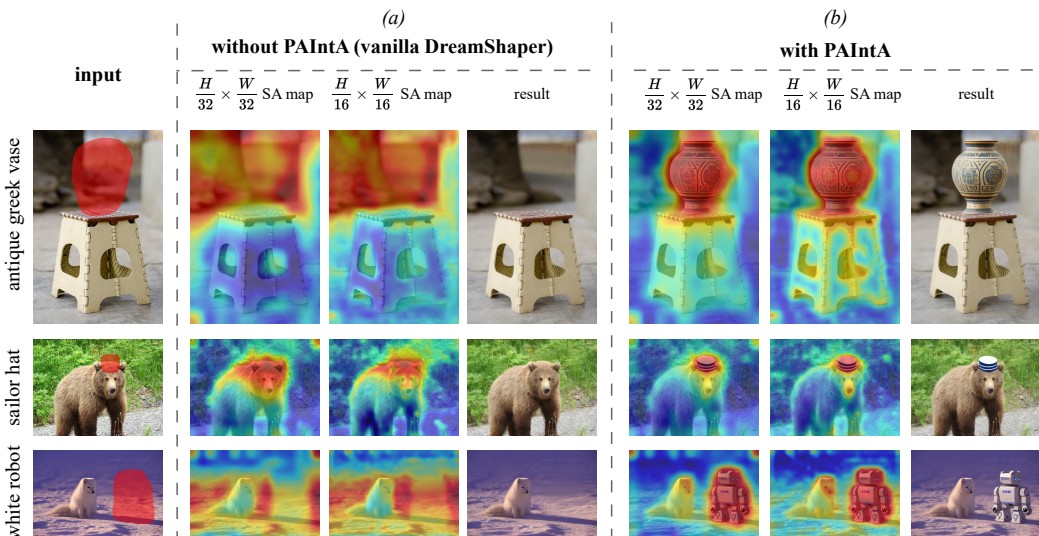

Figure 3: Comparison of self-attention similarity maps averaged across masked pixels for generations without/with PAIntA's scaling of the original self-attention scores. Images are generated from the same seed.

responding cross-attention layers, and $l$ is the number of tokens of the prompt $\tau$. Specifically, we consider CLIP text embeddings of the prompt $\tau$ and separate the ones which correspond to the words of $\tau$ and *End of Text* (EOT) token (in essence we just disregard the SOT token and the null-token embeddings), and denote the set of chosen indices by $ind(\tau) \subset \{1, 2, \ldots, l\}$. We include EOT since (in contrast with SOT) it contains information about the prompt $\tau$ according to the architecture of CLIP text encoder. For each $j^{th}$ pixel we define its similarity with the prompt $\tau$ by summing up it's similarity scores with the embeddings indexed from $ind(\tau)$, i.e. $c_j = \sum_{k \in ind(\tau)} (S_{cross})_{jk}$. Also, we found beneficial to normalize the scores $c_j = clip\left(\frac{c_j - median(c_k; k=1,\ldots,hw)}{max(c_k; k=1,\ldots,hw)}, 0, 1\right)$, where $clip$ is the clipping operation between $[0, 1]$.

Note that in vanilla SD cross-attention layers come after self-attention layers, hence in PAIntA to get query and key tensors $Q_c, K_c$ we borrow the projection layer weights from the next cross-attention module (see fig. 2). Finally we get the output of the PAIntA layer with the residual connection with the input: $Out = X + SoftMax(\tilde{A}_{self}) \cdot V_s$.

### 3.4 REWEIGHTING ATTENTION SCORE GUIDANCE (RASG)

The experiments show that while PAIntA improves prompt-alignment in generation by manipulating the self-attention layers, the issue of prompt-alignment is not completely resolved. Therefore we additionally leverage the concept of post-hoc sampling guidance. The idea, introduced in (Dhariwal & Nichol, 2021) as classifier-guidance and generalized further (Rombach et al. (2022), Chefer et al. (2023), Epstein et al. (2023)), is to guide the sampling (denoising) process $x_t \to x_{t-1}$ to the direction of more prompt-alignment. However, the vanilla post-hoc guidance, as noticed in (Chefer et al., 2023), may shift the domain of diffusion latents $x_{t-1}$ resulting in image quality degradations. To this end we introduce the *Reweighting Attention Score Guidance (RASG)* strategy which benefits from the general DDIM backward process, eq. (3), and introduces a gradient reweighting mechanism resulting in latent domain preservation. Below we first discuss the vanilla post-hoc guidance with an objective function $S(x)$ which we define later to be specifically in charge of prompt-alignment. Then we discuss RASG and how it approaches to the out-of-domain latent shift problem of the vanilla guidance. And finally we present our choice for the post-hoc objective function $S(x)$.

Let $S(x)$ be an objective function the post-hoc guidance mechanism should be applied with. Then, according to (Dhariwal & Nichol, 2021), the update rule[1] for $\epsilon_\theta^t(x_t)$ will be

$$\hat{\epsilon}_\theta^t(x_t) \leftarrow \epsilon_\theta^t(x_t) + \sqrt{1 - \alpha_t} \cdot s \nabla_{x_t} S(x_t), \tag{6}$$

---

[1]for brevity we write $\epsilon_\theta^t(x_t)$ instead of $\epsilon_\theta^t([x_t, \mathcal{E}(I^M), down(M)], \tau)$

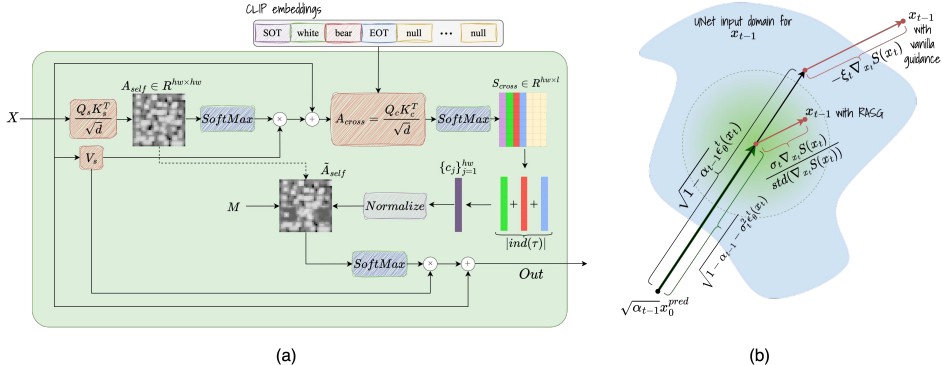

(a)  (b)

Figure 4: (a) PAIntA block takes an input tensor $X \in \mathbb{R}^{h \times w \times d}$ and the CLIP embeddings of $\tau$. After computing the self- and cross-attention scores $A_{self}$ and $A_{cross}$, we update the former (eq. (5)) by scaling with the normalized values $\{c_j\}_{j=1}^{hw}$ obtained from $S_{cross} = SoftMax(A_{cross})$. Finally the the updated attention scores $\tilde{A}_{self}$ are used for the convex combination of the values $V_s$ to get the residual of PAIntA's output. (b) RASG mechanism takes the predicted scaled denoised latent $\sqrt{\alpha_{t-1}}x_0^{pred} = \frac{\sqrt{\alpha_{t-1}}}{\sqrt{\alpha_t}}\left(x_t - \sqrt{1-\alpha_t}\epsilon_\theta(x_t)\right)$ and guides the $x_{t-1}$ estimation process towards minimization of $S(x_t)$ defined by eq. (10). Gradient reweighting makes the gradient term close to being sampled from $\mathcal{N}(\mathbf{0}, \mathbf{1})$ (green area) by so ensuring the domain preservation (blue area).

where $s$ is a hyperparameter controlling the amount of the guidance. This, according to the (deterministic) DDIM process eq. (4), will result in (by substituting $\epsilon_\theta^t(x_t)$ with $\hat{\epsilon}_\theta^t(x_t)$) the following update rule for $x_{t-1}$:

$$x_{t-1} = \sqrt{\alpha_{t-1}}\frac{x_t - \sqrt{1-\alpha_t}\epsilon_\theta^t(x_t)}{\sqrt{\alpha_t}} + \sqrt{1-\alpha_{t-1}}\epsilon_\theta^t(x_t) - \xi_t\nabla_{x_t}S(x_t),$$
$$\xi_t = \sqrt{1-\alpha_t} \cdot s\left(\frac{\sqrt{1-\alpha_t}\sqrt{\alpha_{t-1}}}{\sqrt{\alpha_t}} - \sqrt{1-\alpha_{t-1}}\right). \tag{7}$$

Notice that in eq. (7) we get the additional term $-\xi_t\nabla_{x_t}S(x_t)$ not present in the original sampling process given by eq. (4). This term may shift the original distribution of $x_{t-1}$.

To this end we introduce the *Reweighting Attention Score Guidance (RASG)* strategy which benefits from the general DDIM backward process (eq. (3)) and introduces a gradient reweighting mechanism resulting in latent domain preservation. Specifically, according to Claim 1, $x_{t-1}$ obtained either by eq. (4) or by eq. (3) remains in the required domain (see fig. 4). Hence in eq. (3) by replacing the stochastic component $\epsilon_t$ by the rescaled version of the gradient $\nabla_{x_t}S(x_t)$ (to make it closer to a sampling from $\mathcal{N}(\mathbf{0}, \mathbf{1})$), we can keep $x_{t-1}$ in the required domain and at the same time guide its sampling towards minimization of $S(x_t)$. Rescaling of the gradient $\nabla_{x_t}S(x_t)$ is done by dividing it on its standard deviation (we do not change the mean to keep the direction of the $S(x_t)$ minimization, for more discussion see Appendix C). Thus, RASG sampling is done by the formula

$$x_{t-1} = \sqrt{\alpha_{t-1}}\frac{x_t - \sqrt{1-\alpha_t}\epsilon_\theta^t(x_t)}{\sqrt{\alpha_t}} + \sqrt{1-\alpha_{t-1} - \sigma_t^2}\epsilon_\theta^t(x_t) + \sigma_t\frac{\nabla_{x_t}S(x_t)}{\text{std}(\nabla_{x_t}S(x_t))}. \tag{8}$$

Now let us define the function $S(x)$ (for more discussion on its choice see Appendix C). We want the objective $S(x)$ to guide the sampling to more prompt-alignment, at the same time keeping the generated object in the mask-indicated region. On the other hand we know that (Chefer et al., 2023) cross-attention maps contain similarities between the prompt and the pixels of latent $x_t$, therefore the cross-attention maps can serve as a segmentation maps for the objects described via the given prompt. Thus we define $S(x)$ as a segmentation loss function between the cross-attention maps and the given inpainting mask. Below we discuss the construction of $S(x)$ in detail.

First we consider all cross-attention maps $A_{cross}$ with the output resolution of $\frac{H}{32} \times \frac{W}{32}$: $A_{cross}^1, \ldots, A_{cross}^m \in \mathbb{R}^{(H/32 \cdot W/32) \times l}$, where $m$ is the number of such cross-attention layers, and

$l$ is the number of token embeddings. Then for each $k \in ind(\tau) \subset \{1, \ldots, l\}$ we average the attention maps and reshape to $\frac{H}{32} \times \frac{W}{32}$:

$$\overline{A}^k_{cross}(x_t) = \frac{1}{m} \sum_{i=1}^{m} A^i_{cross}[:, k] \in \mathbb{R}^{\frac{H}{32} \times \frac{W}{32}}. \tag{9}$$

Using post-hoc guidance with $S(x_t)$ we aim to maximize the attention scores in the unknown region determined by the binary mask $M \in \{0, 1\}^{\frac{H}{32} \times \frac{W}{32}}$, hence we take the average negative binary cross entropy between $\overline{A}^k(x_t)$ and $M$ ($M$ is downscaled with NN interpolation, $\sigma$ here is sigmoid):

$$S(x_t) = \sum_{k \in ind(\tau)} \sum_{i=1}^{\frac{H}{32} \cdot \frac{W}{32}} [M_i \log \sigma(\overline{A}^k_{cross}(x_t)_i) + (1 - M_i) \log(1 - \sigma(\overline{A}^k_{cross}(x_t)_i))]. \tag{10}$$

## 4 EXPERIMENTS

### 4.1 IMPLEMENTATION DETAILS

We apply HD-Painter on 3 different Stable Diffusion models: Stable Diffusion 1.5, Stable Diffusion 2.0 and Dreamshaper-8 (Lykon, 2023). PAIntA is used to replace the self attention layers on the $H/32 \times W/32$ and $H/16 \times W/16$ resolutions for the first half of generation steps. For RASG we select only cross-attention similarity matrices of the $H/32 \times W/32$ resolution since utilizing higher resolutions did not offer significant improvements. For hyperparameters $\{\sigma_t\}_{t=1}^{T}$ we chose

$$\sigma_t = \eta \sqrt{(1 - \alpha_{t-1})/(1 - \alpha_t)} \sqrt{1 - \alpha_t/\alpha_{t-1}}, \; \eta = 0.15. \tag{11}$$

In PAIntA's implementation, we reuse calculated cross-attention similarity maps, which results in a very small performance impact. With PAIntA the model is about just 10 % slower, making $\sim 3.3$ seconds from $\sim 3$ seconds of the baseline.

For RASG, naturally, the backward pass of the model increases the runtime about twice. However, optimizations, like using RASG only for a subset of steps, etc., can potentially greatly decrease the runtime while keeping the generation distribution. We keep such investigations for future research.

### 4.2 EXPERIMENTAL SETUP

Here we compare with existing state-of-the-art methods such as GLIDE (Nichol et al., 2021), Stable 2.0 Inpainting (Rombach et al., 2022), DreamShaper Inpainting (Lykon, 2023), Blended Latent Diffusion (BLD) (Avrahami et al., 2023), ControlNet-Inpainting (Zhang & Agrawala, 2023) (with DreamShaper base), SDXL-Inpainting and SmartBrush (Xie et al., 2023). As authors of the Smart-Brush paper don't provide code and model, we reproduce it according to paper and refer to it as *SmartBrush reprod.*. We present the results of SmartBrush reprod. based on DreamShaper text-to-image model, since it had the best performance. We evaluate the methods on a random sample of 10000 (image, mask, prompt) triplets from the validation set of MSCOCO 2017 (Lin et al., 2014), where the prompt is chosen as the label of the selected instance mask. We noticed that when a precise mask of a recognizable shape is given to Stable Inpainting, it tends to ignore the prompt and inpaint based on the shape. To prevent this, we use the convex hulls of the object segmentation masks and compute the metrics accordingly.

We evaluate the CLIP score on a cropped region of the image using the bounding box of the input mask. As CLIP score can still assign high scores to adversarial examples, we additionally compute the generation class accuracy. So, we utilize a pre-trained instance detection model for MSCOCO: MMDetection (Chen et al., 2019). We run it on the cropped area of the generated image, and, as there might be more than one objects included in the crop, we treat the example as positive if the prompt label is in the detected object list.

To measure the visual fidelity of the results we employ the LAION aesthetic score.[2] The aesthetic score is computed by an MLP trained on 5000 image-rating pairs from the Simulacra Aesthetic Captions dataset (Pressman et al., 2022), and can be used to assign a value from the $[0, 10]$ range to images based on their aesthetic appeal.

---

[2]https://github.com/christophschuhmann/improved-aesthetic-predictor

Table 1: Quantitative comparison. 95% confidence interval of 5 runs with different seeds.

| Model Name | CLIP score ↑ | Accuracy, % ↑ | Aesthetic score ↑ |
|---|---|---|---|
| GLIDE | 25.09 ± 0.01 | 43.08 ± 0.30 | 4.476 ± 0.002 |
| BLD | 25.64 ± 0.05 | 55.64 ± 0.59 | 4.822 ± 0.006 |
| SDXL Inpainting | 24.80 ± 0.02 | 52.98 ± 0.91 | 4.682 ± 0.024 |
| DreamShaper-ControlNet Inp. | 25.73 ± 0.01 | 58.74 ± 0.27 | 4.946 ± 0.005 |
| SmartBrush reprod. | 25.86 ± 0.03 | 66.88 ± 0.48 | 4.856 ± 0.004 |
| Stable 1.5 Inpainting | 25.10 ± 0.02 | 55.25 ± 0.46 | 4.881 ± 0.006 |
| Stable 2.0 Inpainting | 25.07 ± 0.03 | 51.74 ± 0.54 | 4.885 ± 0.006 |
| DreamShaper Inpainting | 25.61 ± 0.02 | 58.93 ± 0.18 | 4.965 ± 0.004 |
| Stable 1.5 + HD-Painter | 25.83 ± 0.05 | 59.57 ± 0.58 | 4.864 ± 0.006 |
| Stable 2.0 + HD-Painter | **26.48** ± 0.03 | 59.74 ± 0.56 | 4.846 ± 0.011 |
| Dreamshaper 8 + HD-Painter | 26.32 ± 0.03 | **68.05** ± 0.48 | **4.980** ± 0.003 |

### 4.3 QUANTITATIVE AND QUALITATIVE ANALYSIS

Table 1 shows that HD-Painter increases the prompt alignment of the corresponding baseline models. It can be noticed that while SmartBrush trained over DreamShaper Inpainting improves the accuracy over the baseline, the CLIP score improvement is marginal and the overall quality is significantly dropped according to aesthetic score. On the other hand, our method significantly improves the prompt-alignment as measured by both CLIP score and accuracy while also maintaining the quality. Additionally, we performed a user study (see Appendix Appendix D).

The examples in fig. 5 demonstrate qualitative comparison between our method and the other state-of-the-art approaches. In many cases the baseline DreamShaper Inp. generates a background (fig. 5, columns 1, 3, 5) or reconstructs the missing regions as continuation of the known region objects disregarding the prompt (fig. 5, columns 4, 6, 7), while our method, thanks to the combination of PAIntA and RASG, successfully generates the target objects. Notice that even though DreamShaper-ControlNet-Inpainting and SmartBrush reprod. may also generate the required object, the quality of the generation is poor compared to ours.

Table 2: Ablation for PAIntA and RASG on the Dreamshaper 8 base. 95% confidence interval of 5 runs with different seeds.

| Model Name | CLIP score ↑ | Accuracy ↑ | Aesthetic score ↑ |
|---|---|---|---|
| base (DreamShaper Inp.) | 25.61 ± 0.02 | 58.93 ± 0.18 | 4.965 ± 0.004 |
| only PAIntA | 26.07 ± 0.03 | 63.95 ± 0.50 | **4.985 ± 0.003** |
| only RASG | 25.94 ± 0.02 | 63.75 ± 0.48 | 4.965 ± 0.003 |
| RASG & PAIntA | **26.32 ± 0.03** | **68.05 ± 0.48** | 4.980 ± 0.003 |

### 4.4 ABLATION STUDY

In table 2 we show that PAIntA and RASG separately provide substantial improvements to the model quantitatively. We also provide more discussion on each of them in our supplementary material, including thorough analyses on their impact, demonstrated by visuals. For qualitative ablation study see figs. 6 and 7 in Appendices B and C.

## 5 CONCLUSION

In this paper, we introduced a training-free method to prompt-faithful text-guided image inpainting, addressing the prevalent challenges of prompt neglect: background and nearby object dominance. Our contributions, the Prompt-Aware Introverted Attention (PAIntA) layer and the Reweighting Attention Score Guidance (RASG) mechanism, effectively mitigate the mentioned issues leading our method to surpass the existing state-of-the-art approaches qualitatively and quantitatively.

## 6 REPRODUCIBILITY STATEMENT

We have included the codebase, evaluation scripts, and comprehensive instructions for reproducing our experiments in the supplementary material. This ensures the reproducibility of our results and facilitates independent validation. Furthermore, we report confidence intervals for all computed metrics, demonstrating the robustness of our findings to random factors.

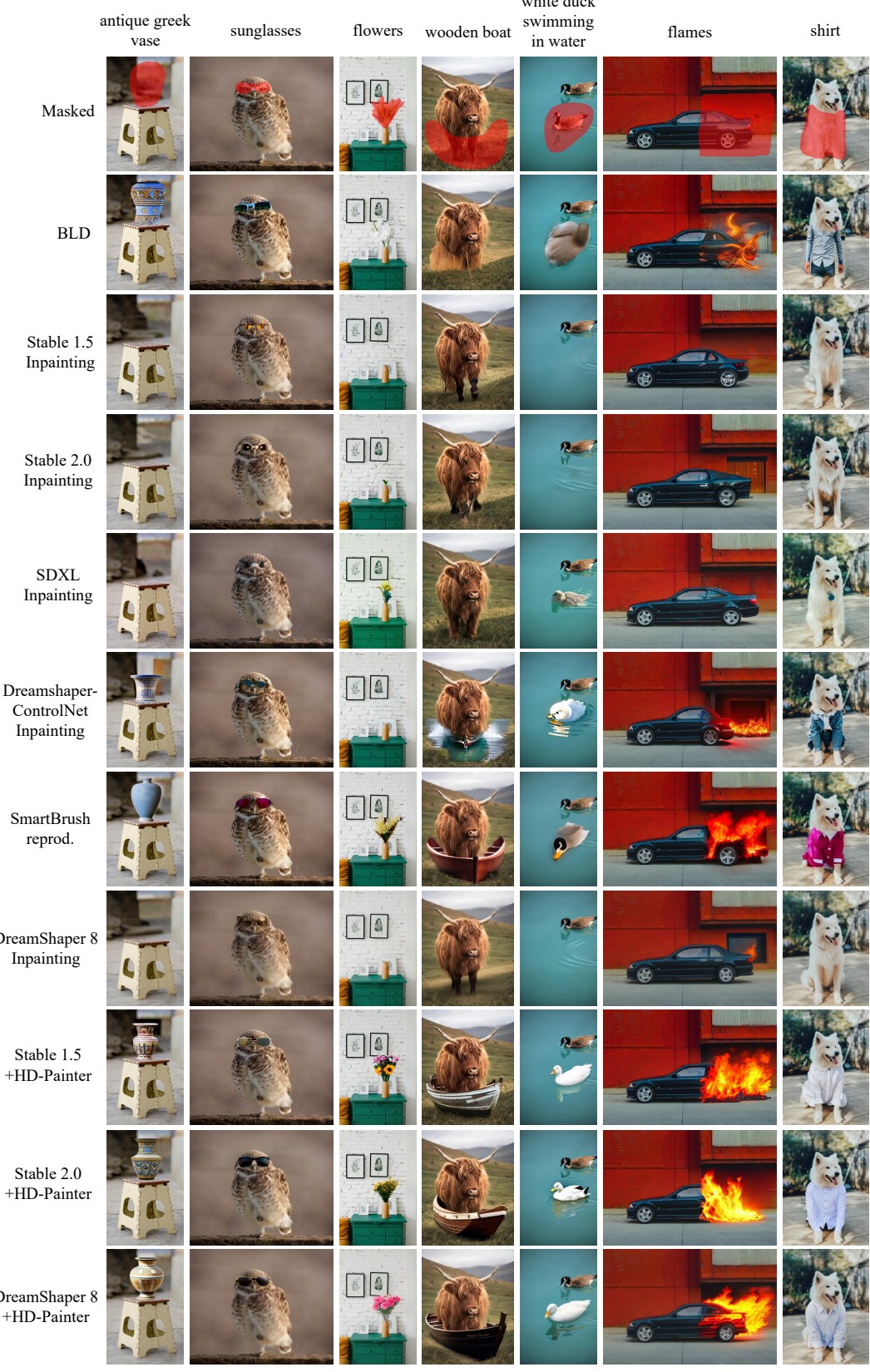

Figure 5: Comparison with state-of-the-art text-guided inpainting methods. Zoom in for details. For more comparison see Appendix A.

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

## A    EXTENDED QUALITATIVE COMPARISON

In fig. 18 we show more visual comparison with the other state-of-the-art methods. Figure 19 includes more comparison on the validation set of MSCOCO 2017 (Lin et al., 2014). The results show the advantage of our method over the baselines.

## B    DISCUSSION ON PAINTA

In this section we discuss the effectiveness of the proposed PAIntA module as a plug-in replacement for self-attention (SA) layers. To that end, first we visualize SA similarity maps averaged across masked locations from resolutions $H/16 \times W/16$ and $H/32 \times W/32$ where PAIntA is applied (see fig. 3). Then, we see that PAIntA successfully scales down the similarities of masked locations with prompt-unrelated locations from the known region, and, as a result, a prompt-specified object is generated inside the mask.

For a given resolution ($H/16 \times W/16$ or $H/32 \times W/32$), in order to visualize the average SA similarity map across masked pixels, first we resize the input mask to match the dimensions of the corresponding resolution (we use nearest interpolation in resize operation). Then, for each SA layer in the given resolution, we form a 2D similarity map by reshaping and averaging the similarity matrix rows corresponding to the masked region. Further, we average obtained 2D similarity maps across all SA layers (of the given resolution) and diffusion timesteps. More specifically, if $A_{self}^1, \ldots, A_{self}^L \in \mathbb{R}^{hw \times hw}$ ($h \times w$ is either $H/16 \times W/16$ or $H/32 \times W/32$) are the self-attention matrices of Stable Inpainting layers of the given resolution, and, respectively, are being updated by PAIntA to the matrices $\tilde{A}^i{}_{self}$ (see eq. (5)), then we consider the following similarity maps:

$$
\begin{aligned}
A &= \frac{1}{|M| \cdot L} \sum_{i, M_i=1} \sum_{l=1}^{L} (A_{\text{self}}^l)_i \in \mathbb{R}^{hw}, \\
\tilde{A} &= \frac{1}{|M| \cdot L} \sum_{i, M_i=1} \sum_{l=1}^{L} (\tilde{A}_{\text{self}}^l)_i \in \mathbb{R}^{hw},
\end{aligned}
\tag{12}
$$

and reshape them to 2D matrices of size $h \times w$. So, $A_{ij}$ and $\tilde{A}_{ij}$ show the average amount in which masked pixels attend to to other locations in the cases of the vanilla self-attention and PAIntA respectively. Finally, in order to visualize the similarity maps, we use bicubic resize operation to match it with the image dimensions and plot the similarity heatmap using JET colormap from OpenCV (Itseez, 2015).

Next, we compare the generation results and corresponding similarity maps obtained from above procedure when PAIntA's SA scaling is (the case of $\tilde{A}$) or is not (the case of $A$) used. Because PAIntA's scaling is only applied on $H/32 \times W/32$ and $H/16 \times W/16$ resolutions, we are interested in those similarity maps. Rows 1-3 in fig. 3 demonstrate visualizations on *nearby object dominance* issue (when known objects are continued to the inpainted region while ignoring the prompt) of the vanilla diffusion inpainting, while rows 4-6 demonstrate those of with *background dominance* issue (when nothing is generated, just the background is coherently filled in).

For example, on row 1 (fig. 3) in case of *Stable Inpainting without PAIntA* generation, the average similarity of the masked region is dominated by the known regions of the car on both 16 and 32 resolutions. Whereas, as a result of PAIntA scaling application, the average similarity of the masked region with the car is effectively reduced, and the masked region is generated in accordance to the input prompt.

Row 4 (fig. 3) demonstrates an example where the result without PAIntA continues the background based on visual context instead of following the user prompt. In this case, visualization shows that usage of PAIntA successfully reduces the similarity of the masked region with the unrelated background. As a result, by reducing the similarity of masked region with the unrelated known regions PAIntA enables prompt-faithful generation. You can find additional examples of PAIntA's effect on the final generation in fig. 6.

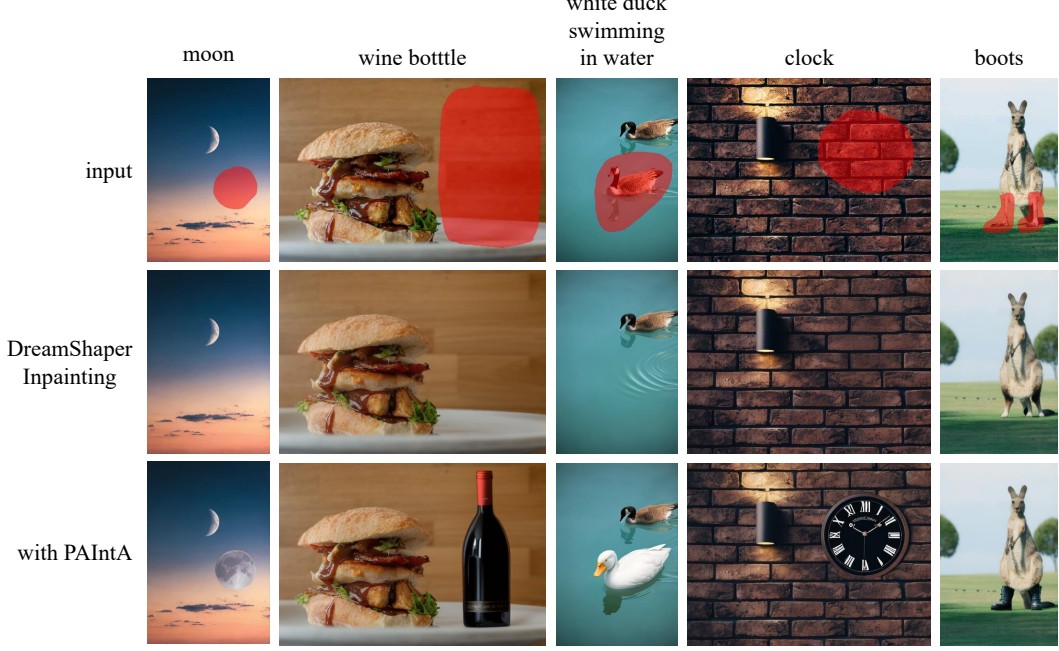

Figure 6: Visual ablation of PAIntA. Generated images use the same seed. In row 3 only PAIntA is used.

## C DISCUSSION ON RASG

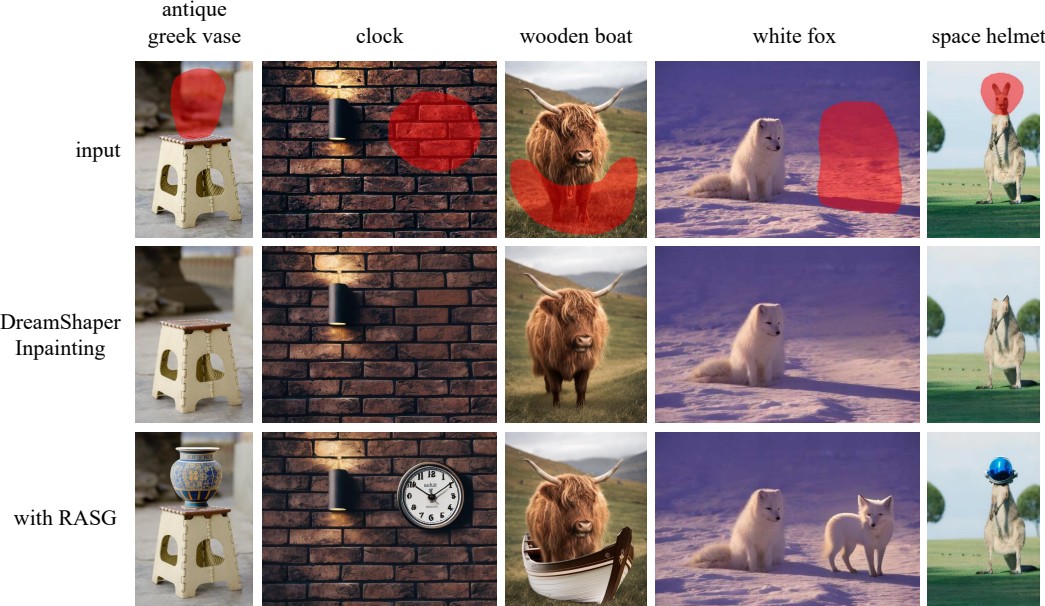

Figure 7: Visual ablation of RASG. Generated images use the same seed. In row 3 only RASG is used.

In this section we discuss the choice of RASG objective guidance function $S(x)$, then demonstrate the effect of RASG and motivate the part of gradient reweighting by its standard deviation. Finally, we present additional examples of RASG's effect on the final generation in fig. 7.

## C.1 THE OBJECTIVE FUNCTION $S(x)$

As we already mentioned in the main paper, Stable Inpainting may fail to generate certain objects in the prompt, completely neglecting them in the process. We categorized these cases into two types, namely background and nearby object dominance issues. Chefer et al. (2023) also mentions these issues but for text-to-image generation task, and refers them as *catastrophic neglect* problem. To alleviate this problem Chefer et al. (2023) propose a mechanism called *generative semantic nursing*, allowing the users to "boost" certain tokens in the prompt, ensuring their generation. In essence the mechanism is a post-hoc guidance with a chosen objective function maximizing the maximal cross-attention score of the image with the token which should be "boosted". This approach can be easily adapted to the inpainting task by just restricting the maximum to be taken in an unknown region so that the object is generated there, and averaging the objectives across all tokens, since we don't have specific tokens to "boost", but rather care about all of them. In other words, by our notations from the main paper, the following guidance objective funciton can be used:

$$S(x_t) = -\frac{1}{|ind(\tau)|} \sum_{k \in ind(\tau)} \max_{i: M_i=1} \{\overline{A}^k(x_t)_i\}. \tag{13}$$

However we noticed that with this approach the shapes/sizes of generated objects might not be sufficiently aligned with the shape/size of the input mask, which is often desirable for text-guided inpainting (see fig. 9). Therefore, we utilize the segmentation property of cross-attention similarity maps, by so using *Binary Cross Entropy* as the energy function for guidance (see eq. (10) in the main paper). As can be noticed from fig. 9 the results with the binary cross-entropy better fit the shape of the inpaining mask.

## C.2 EFFECT OF RASG STRATEGY

Although the objective function $S(x)$ defined by eq. (10) (main paper) results in better mask shape/size aligned inpainting, the vanilla post-hoc guidance may lead the latents to become out of their trained domain as also noted by Chefer et al. (2023): *"many updates of $x_t$ may lead to the latent becoming out-of-distribution, resulting in incoherent images"*. Due to this the post-hoc guidance mechanism (semantic nursing) by Chefer et al. (2023) is done using multiple iterations of very small, iterative perturbations of $x_t$, which makes the process considerably slow. In addition, the generation can still fail if the iterative process exceeds the maximum iteration limit without reaching the necessary thresholds.

Thanks to RASG's seamless integration of the $\nabla_{x_t} S(x_t)$ gradient component into the general form of DDIM diffusion sampling, our RASG mechanism keeps the modified latents $x_t$ within the expected distribution, while introducing large enough perturbations to $x_t$ with only one iteration of guidance per time-step. This allows to generate the objects described in the prompts coherently with the known region without extra-cost of time.

Figure 8 demonstrates the advantage of RASG's strategy over the vanilla guidance mechanism. Indeed, in the vanilla post-hoc guidance there is a hyperparameter $s$ controlling the amount of guidance. When $s$ is too small (e.g. close to 0 or for some cases $s = 100$) the vanilla guidance mechanism does not show much effect due to too small guidance from $s\nabla_{x_t}S(x_t)$. Then with increasing the hyperparameter ($s = 1000, 10000$) one can notice more and more text/shape alignment with prompt/inpainting mask, however the generated results are unnatural and incoherent with the known region. This is made particularly challenging by the fact, that different images, or even different starting seeds with the same input image might require different values of the perturbation strength to achieve the best result. In contrast, RASG approach is *hyperparameter-free* allowing both: prompt/mask-aligned and naturally looking results.

## C.3 RESCALING WITH STANDARD DEVIATION

The core idea of RASG is to automatically scale perturbation using certain heuristics, such that the guidance process has a consistent effect on the output, without harming the quality of the image. Our main heuristic relies on the fact that Song et al. (2021) have defined a parametric family of stochastic denoising processes, which can all be trained using the same training objective as DDPM

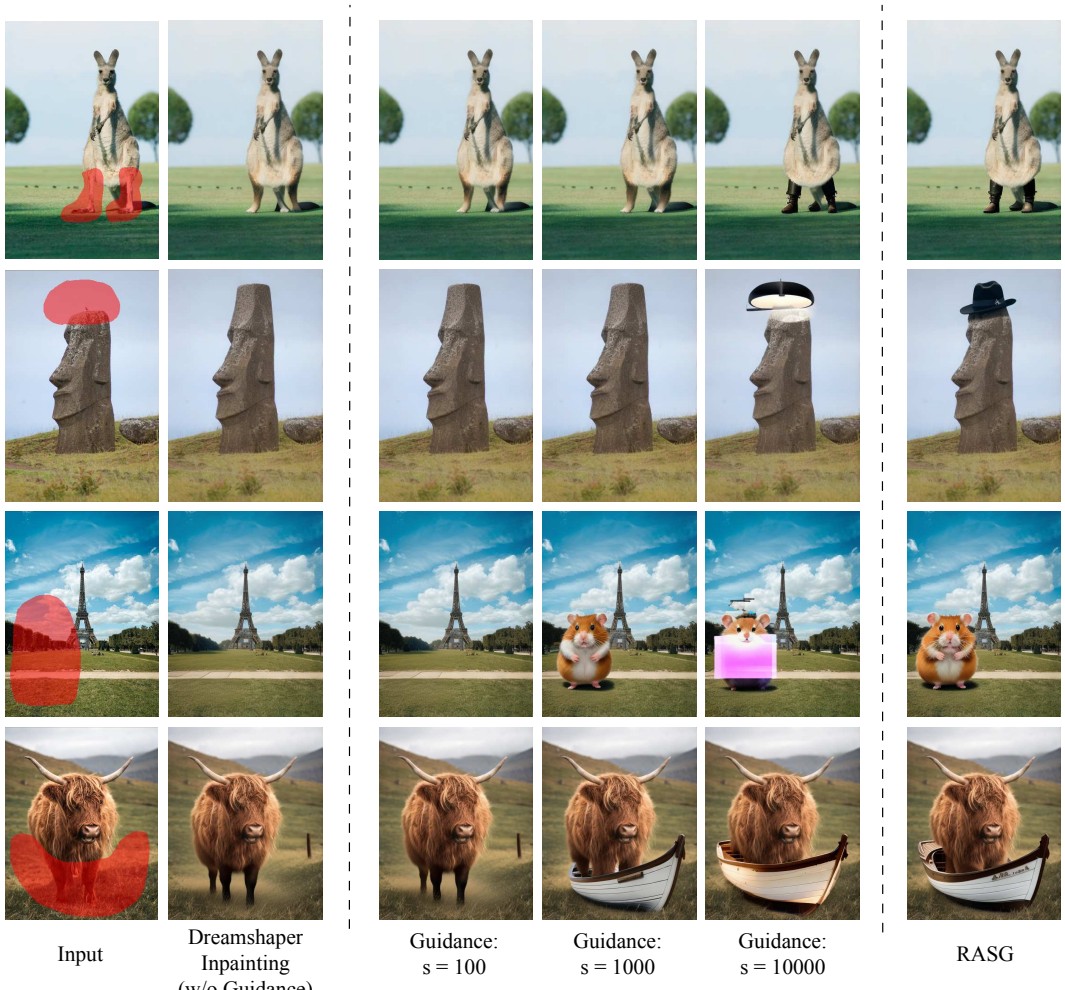

| Input | Dreamshaper Inpainting (w/o Guidance) | Guidance: s = 100 | Guidance: s = 1000 | Guidance: s = 10000 | RASG |
|---|---|---|---|---|---|

Figure 8: Comparison of RASG strategy with default Stable Inpainting and vanilla guidance mechanism with different guidance scales. In contrast to vanilla guidance, where the generation highly depends on the guidance scale, RASG consistently produces naturally looking and prompt-aligned results.

(Ho et al., 2020). Recall the general form of parametric family of DDIM sampling processes:

$$x_{t-1} = \sqrt{\alpha_{t-1}} \frac{x_t - \sqrt{1 - \alpha_t} \epsilon_\theta^t(x_t)}{\sqrt{\alpha_t}} + \sqrt{1 - \alpha_{t-1} - \sigma_t^2} \epsilon_\theta^t(x_t) + \sigma_t \epsilon_t, \qquad (14)$$

where $\epsilon_t \sim \mathcal{N}(\mathbf{0}, \mathbf{1})$. Particularly $\epsilon_t$ can be taken to be collinear with the gradient $\nabla_{x_t} S(x_t)$ which will result in $x_{t-1}$ distribution preservation by at the same time guiding the generation process towards minimization of $S(x_t)$.

Therefore we propose to scale the gradient $\nabla_{x_t} S(x_t)$ with a value $\lambda$ and use instead of $\epsilon_t$ in the general form of DDIM. To determine $\lambda$ we analyse the distribution of $\nabla_{x_t} S(x_t)$ and found out that the values of the gradients have a distribution very close to a gaussian distribution, with $0$ mean and some arbitary $\sigma$, which changes over time-step/image (fig. 10). Therefore, computing the standard deviation of the values of $\nabla_{x_t} S(x_t)$, and normalizing it by $\lambda = \frac{1}{std(\nabla_{x_t} S(x_t))}$ results in the standard normal distribution (see fig. 11). So the final form of RASG guidance strategy is

$$x_{t-1} = \sqrt{\alpha_{t-1}} \frac{x_t - \sqrt{1 - \alpha_t} \epsilon_\theta^t(x_t)}{\sqrt{\alpha_t}} + \sqrt{1 - \alpha_{t-1} - \sigma_t^2} \epsilon_\theta^t(x_t) + \sigma_t \frac{\nabla_{x_t} S(x_t)}{std(\nabla_{x_t} S(x_t))}. \qquad (15)$$

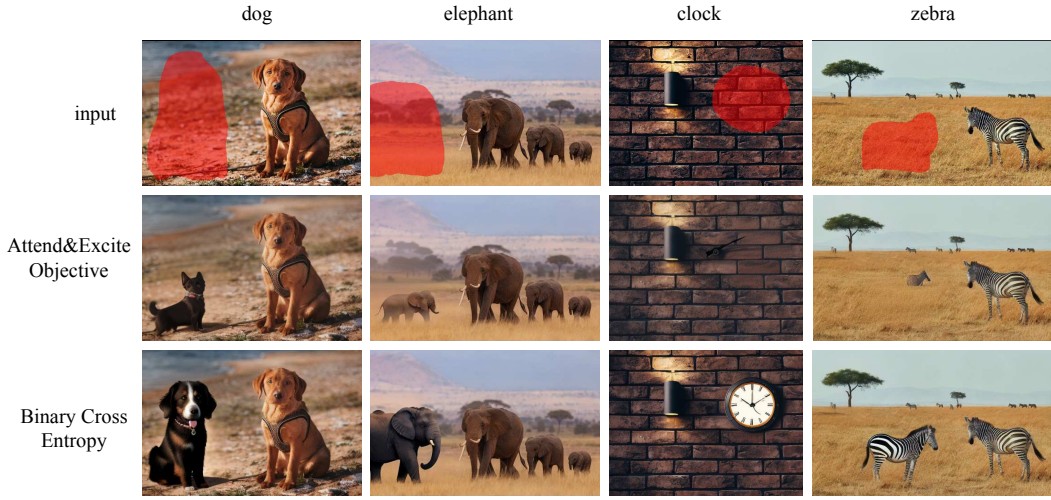

Figure 9: Comparison of the Binary Cross Entropy engery function to modifed version of Attend & Excite. Images generated from the same seed.

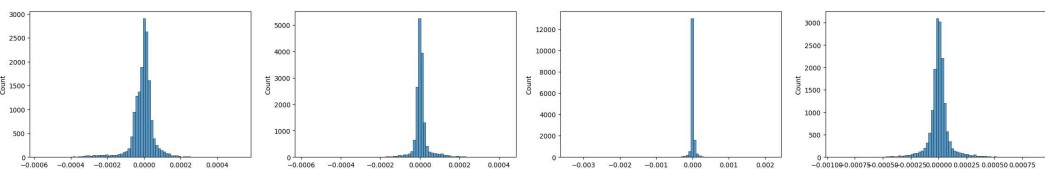

Figure 10: Histogram of $\nabla_{x_t} S(x_t)$ values (i.e. before gradient standardization)

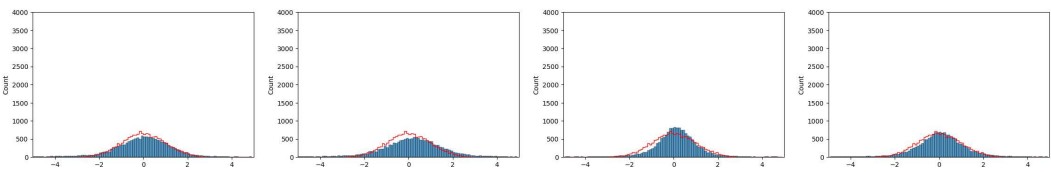

Figure 11: Histogram of $\frac{\nabla_{x_t} S(x_t)}{std(\nabla_{x_t} S(x_t))}$ values (i.e. after gradient standardization)

## D    USER STUDY

We perform a user study for a qualitative comparison with the competitor state-of-the art methods. The 12 participants were shown 20 *(image, mask, prompt)* triplets and the inpainting results of all methods in random order. For each sample image we asked to select the best results based on (*i*) *prompt alignment* and (*ii*) *overall quality*, allowing the choice of no methods when all methods were bad, or multiple methods when the quality was similar. We calculate the total votes for all methods for each question. The results are presented in fig. 12 demonstrating a clear advantage of our method in both aspects over all competitor methods.

## E    INPAINTING-SPECIALIZED CONDITIONAL SUPER-RESOLUTION

Here we discuss our method for high-resolution inpainting utilizing a pre-trained diffusion-based super-resolution model. We leverage the fine-grained information from the known region to upscale the inpainted region (see fig. 13.). In our experiments we utilized Stable Diffusion x4 Upscaler (Rombach et al., 2022).

Assume that $I \in \mathbb{R}^{H \times W \times 3}$ is the high-resolution image we want to inpaint, $M \in \mathbb{R}^{H \times W}$ is the inpainting mask, and $\mathcal{E}$ is the encoder of VQ-GAN (Esser et al., 2021). We start by downscaling

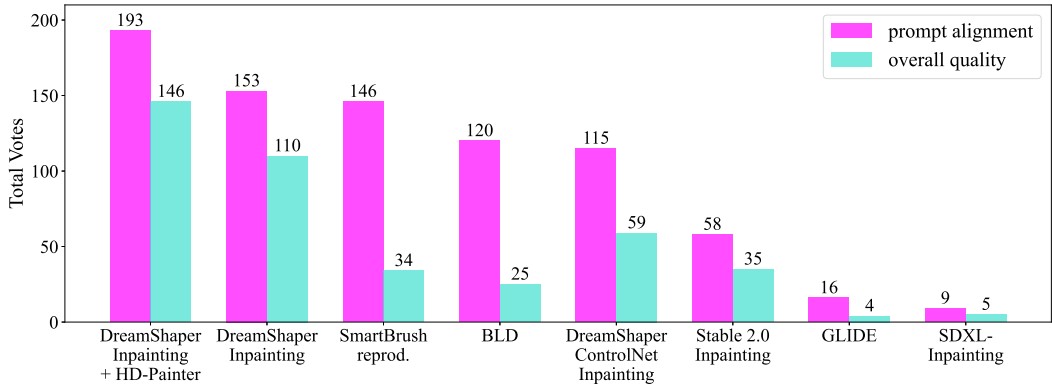

Figure 12: Total votes of each method based on our user study for *prompt alignment* and *overall quality*. Our method HD-Painter has a clear advantage over all competitors.

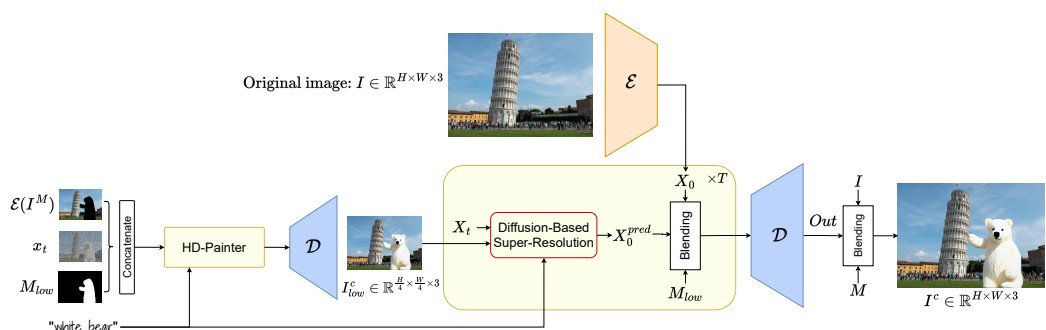

Figure 13: For inpainting-specific super resolution we condition the high-resolution latent $X_t$ denoising process by the lower resolution inpainted result $I_{low}^c$, followed by blending $X_0^{pred} \odot M_{low} + \mathcal{E}(I) \odot (1 - M_{low})$. Finally we get $I^c$ by Poisson blending the decoded output with the original image $I$.

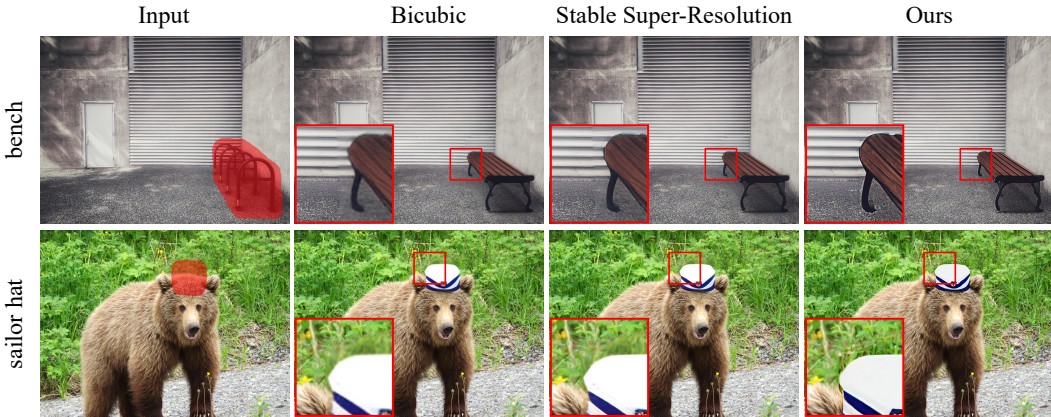

Figure 14: Comparison of our inpainting-specialized super-resolution approach with vanilla upscaling methods for inpainting. Best viewed when zoomed in.

$I$ and $M$ to smaller resolution image $I_{low} \in \mathbb{R}^{\frac{H}{4} \times \frac{W}{4} \times 3}$, and mask $M_{low} \in \mathbb{R}^{\frac{H}{4} \times \frac{W}{4}}$. We then use HD-Painter to obtain a low-resolution inpainted image $I_{low}^c$. To perform inpainting-specialized conditional super-resolution, we consider $X_0 = \mathcal{E}(I)$ and take a standard Gaussian noise $X_T \in \mathbb{R}^{\frac{H}{4} \times \frac{W}{4} \times 4}$. Then we apply a backward diffusion process (eq. (4)) on $X_T$ by using the upscale-

specialized SD model and conditioning it on $I_{low}^c$. After each diffusion step we blend the estimated denoised latent $X_0^{pred} = (X_t - \sqrt{1 - \alpha_t}\epsilon_\theta^t(X_t))/\sqrt{\alpha_t}$ with $X_0$ by using $M_{low}$:

$$X_0^{pred} \leftarrow M_{low} \odot X_0^{pred} + (1 - M_{low}) \odot X_0, \qquad (16)$$

and use the new $X_0^{pred}$ to determine the latent $X_{t-1}$ (by eq. (4)). After the last diffusion step $X_0^{pred}$ is decoded and blended (Poisson blending) with the original image $I$.

It's worth noting that our blending approach is inspired by seminal works (Sohl-Dickstein et al., 2015; Avrahami et al., 2022) blending $X_t$ with the noisy latents of the forward diffusion. However, in contrast to those works, we blend high-frequencies from $X_0$ with the denoised prediction $X_0^{pred}$ allowing noise-free image details propagate from the known region to the missing one during all diffusion steps.

In fig. 14 we compare our inpainting-specialized super-resolution method with vanilla approaches of Bicubic or Stable Super-Resolution-based upscaling of the inpainting results followed by Poisson blending in the unknown region. We can clearly see that our method, leveraging the known region fine-grained information, can seamlessly fill in with high quality.

In figs. 20 and 21 we show more visual comparisons between our method and the approach of Stable Super-Resolution.

## F  LARGE QUANTITY OF OBJECTS

In this section we examine the challenging case, when multiple instances of the target object are already present in the known region of the input image. Many inpainting methods often do not generate the target object in this case when it is already present in the image. Usually this means that the input prompt is considered in the global scope of the image and not just the masked area. Increasing the number of the existing objects makes this problem particularly challenging.

fig. 15 shows the effect of applying HD-Painter on the Dreamshaper-8 base. As evident from the example, HD-Painter enforces that the requested object is generated inside of the given mask, thus ensuring that the object is properly generated.

## G  MULTI-OBJECT MASKS

In this section we evaluate the ability of our method to handle masks consisting of multiple parts. This can be useful when trying to generate multiple objects at the same time. The results presented in fig. 16 show that our method is able to generate objects in the case of multiple masks. Note that this is more relevant when all objects are of the same type, as generating objects of multiple types also requires specifying which type of object should be generated in each region. However, the task we investigate in this paper is text-guided image inpainting, which assumes that the only inputs are the mask and the textual prompt.

## H  LIMITATIONS

Although our method improves the prompt-alignment of existing text-guided inpainting approaches, it still has a dependency on the backbone model, hence inherits some quality limitations. Particularly it may generate extra limbs (the elephant in fig. 17 has 5 legs) or illogical appearances (the sheep appears to have two bodies in fig. 17 after the inpainting).

## I  POTENTIAL NEGATIVE IMPACTS

Our research strives to enhance the accuracy of object generation within the scope of text-guided image inpainting. However, it is crucial to acknowledge the potential negative impacts. The technology could be exploited to create deceptive imagery or disseminate misinformation, raising ethical concerns. While our method is training-free and does not introduce new biases, it is imperative to

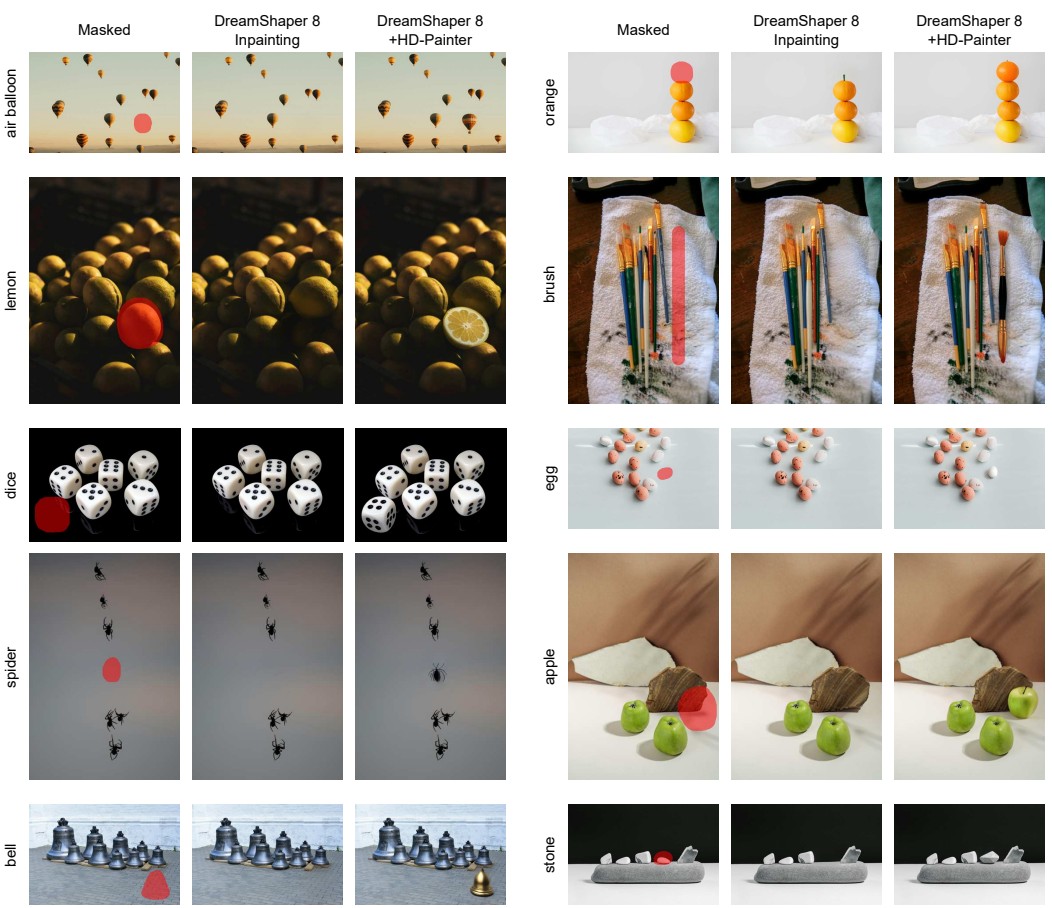

Figure 15: HD-Painter makes sure the target object is generated inside the masked area, even when multiple copies of the object exist outside.

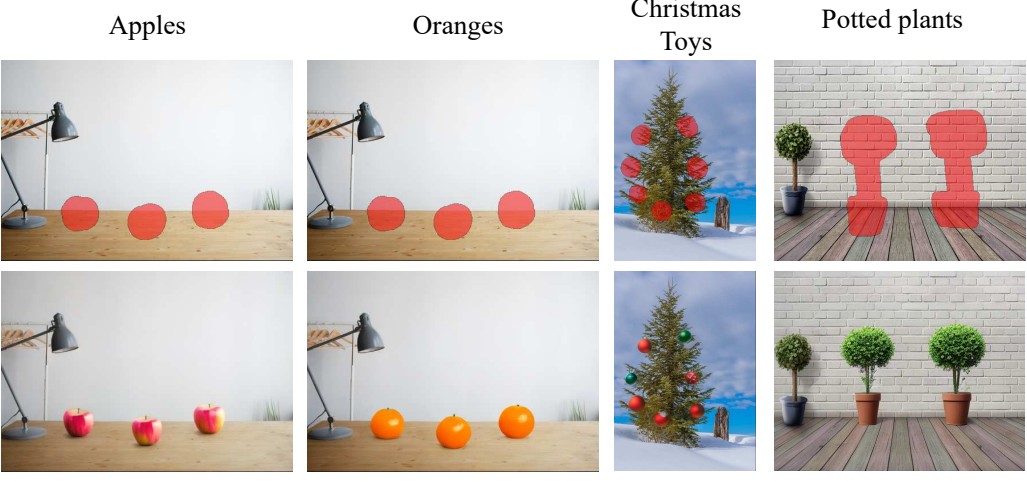

Figure 16: HD-Painter has no trouble generating multiple objects in a single mask

epephant 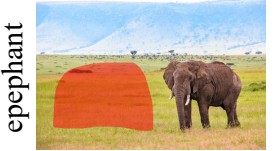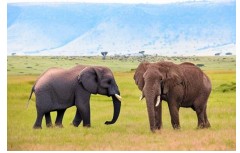   sheep 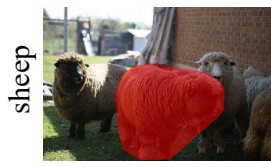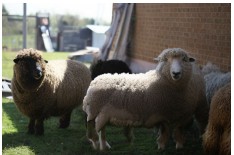

Figure 17: Failure examples produced by our approach.

consider the potential propagation of biases from the base models we build upon. These biases could lead to the generation of content that inadvertently reflects societal or historical prejudices.

To counter these issues, it is essential for the broader research community to establish ethical standards and develop robust methods to detect AI-generated content. Furthermore, efforts should be made to diversify training datasets to reduce inherent biases. While these challenges are significant, the positive implications of our work in areas such as creative arts, design and content creation, when used responsibly, have the potential to surpass the negative repercussions.

## J   MORE EXAMPLES OF OUR METHOD

We present more results of our method both for low-resolution (512 for the long side) images (fig. 22), as well as high-resoltuion (2048 for the long side) (figs. 23 to 25).

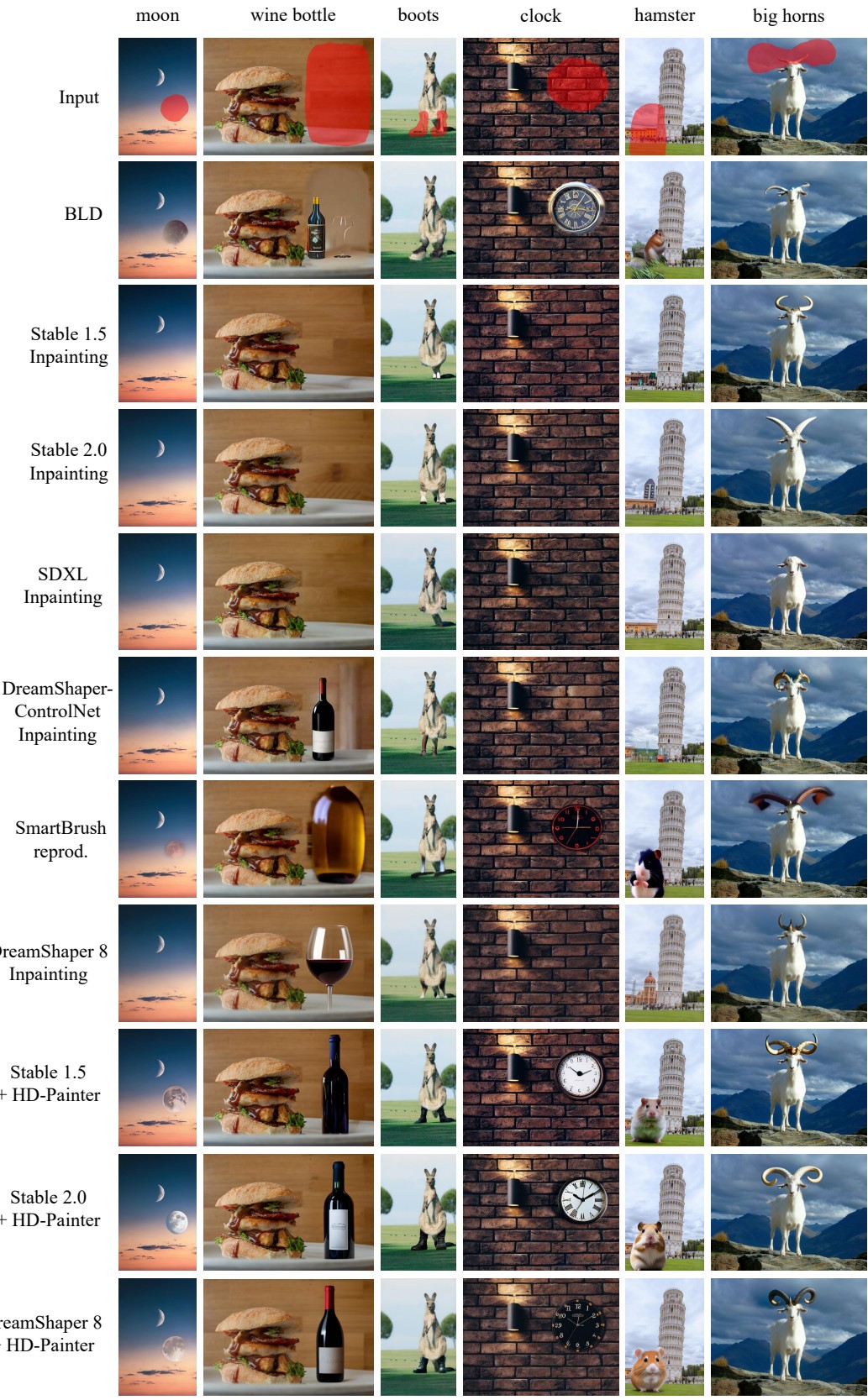

Figure 18: More qualitative comparison results. Zoom in to view the details.

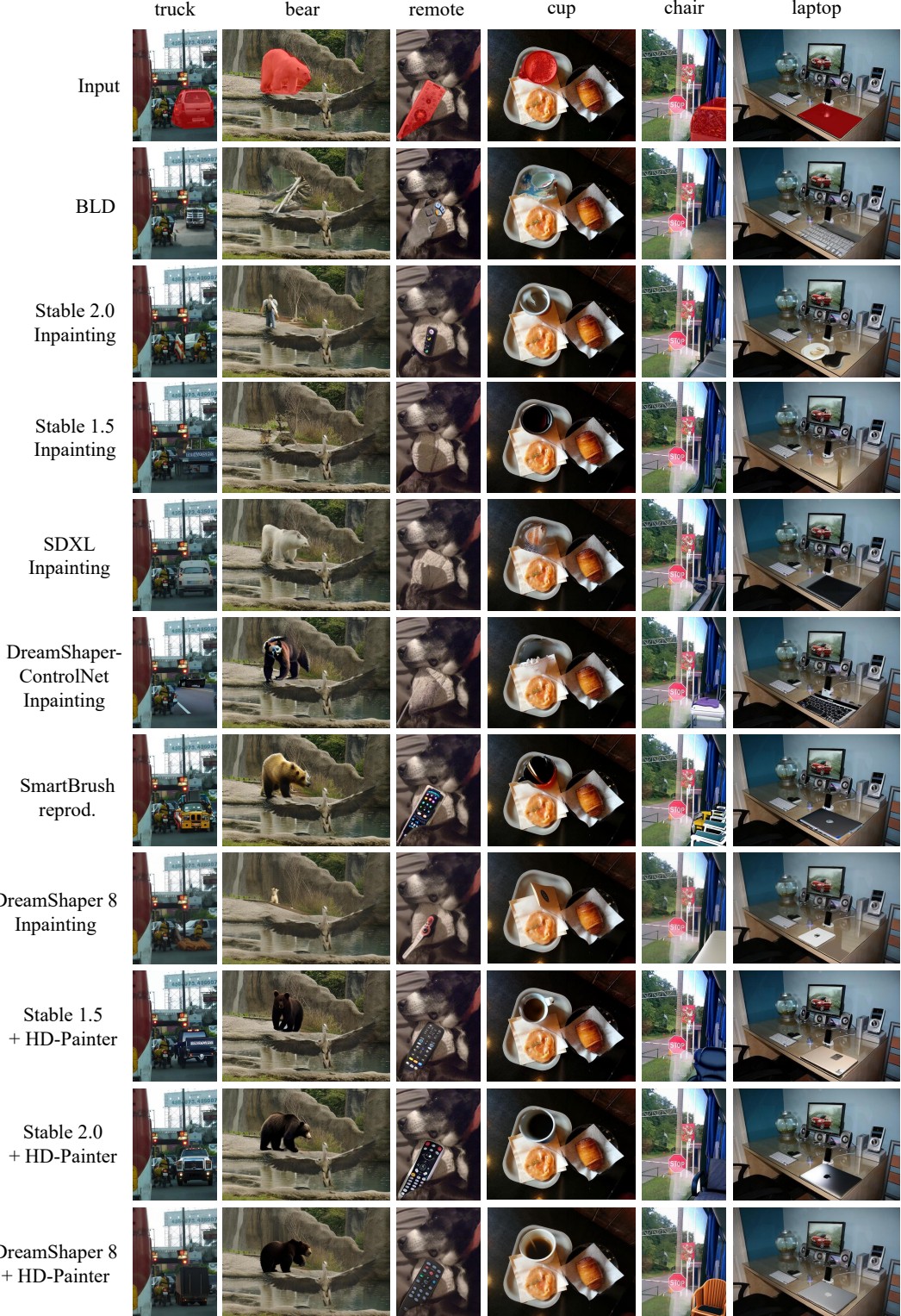

Figure 19: Qualitative comparison results on MSCOCO 2017.

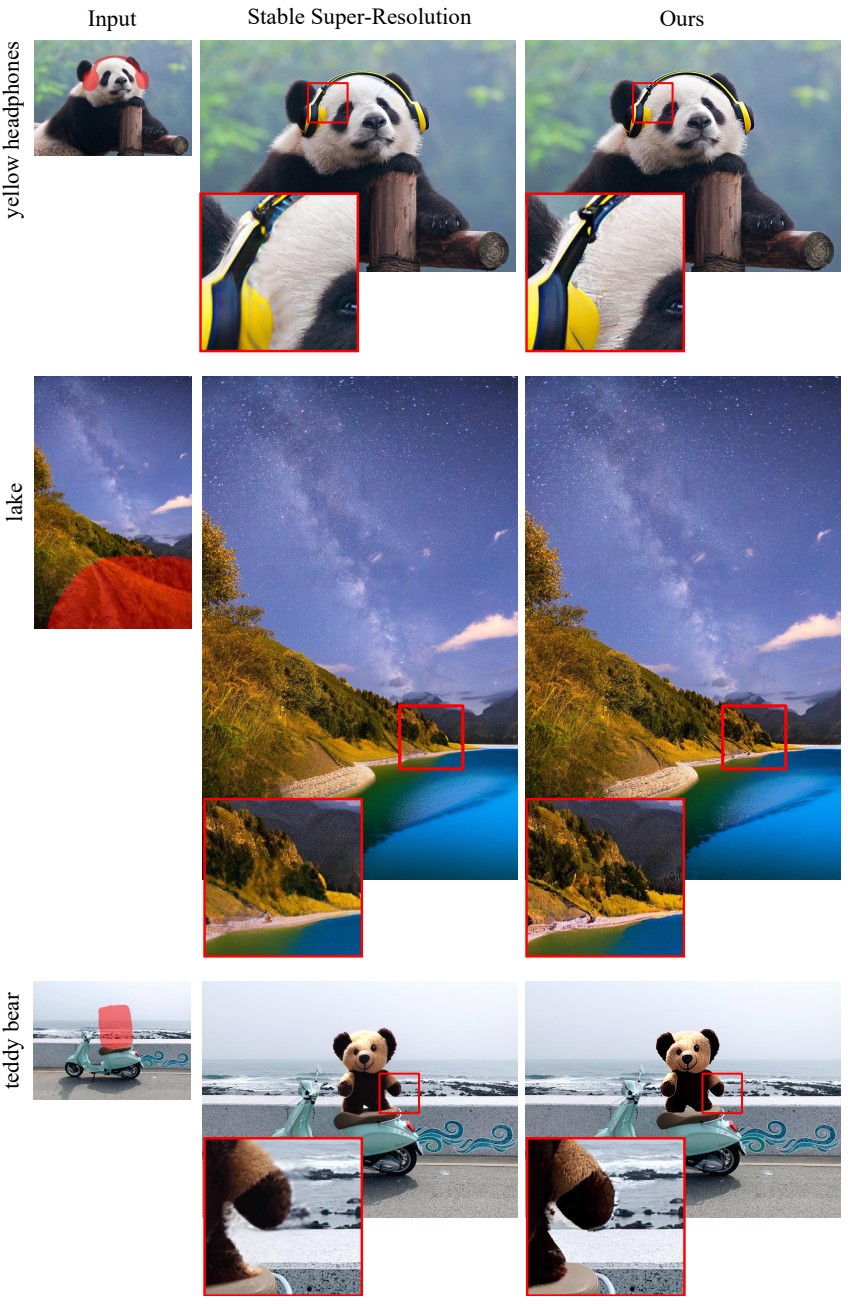

Figure 20: Comparison between vanilla SD 2.0 upscale and our approach. In all examples the large side is 2048px. The cropped region is 256x256px. Best viewed when zoomed in.

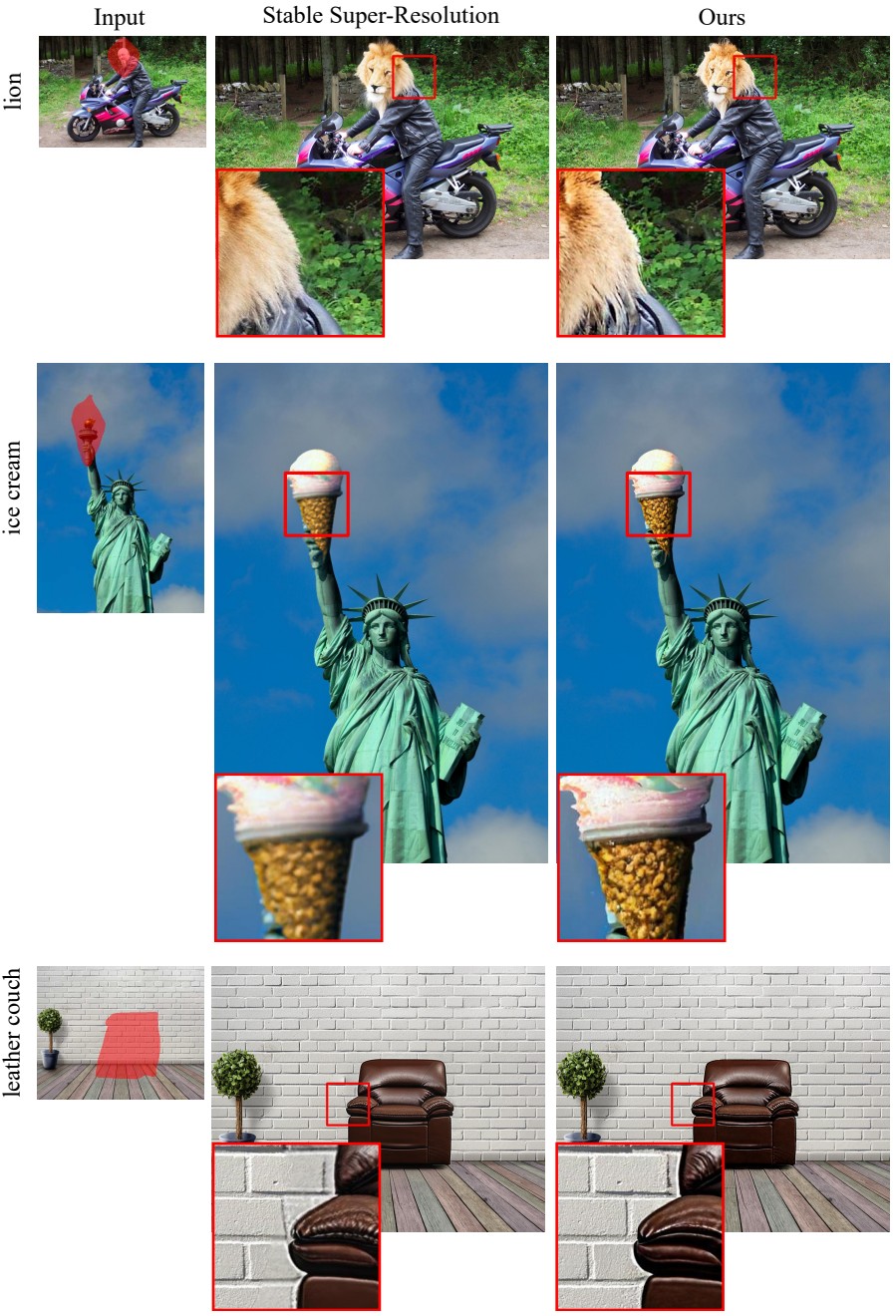

Figure 21: Comparison between vanilla SD 2.0 upscale and our approach. In all examples the large side is 2048px. The cropped region is 256x256px. Best viewed when zoomed in.

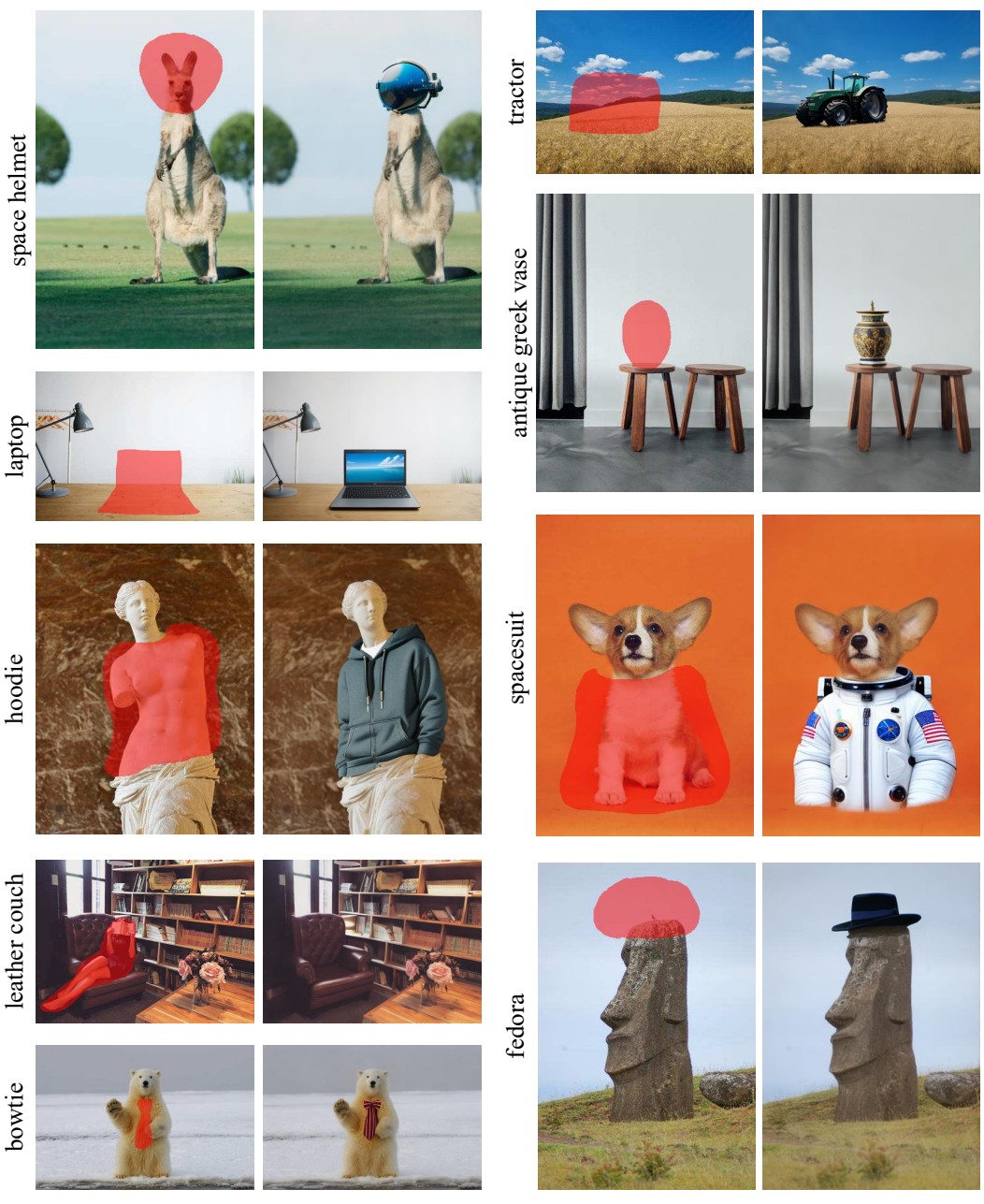

Figure 22: More results of our method.

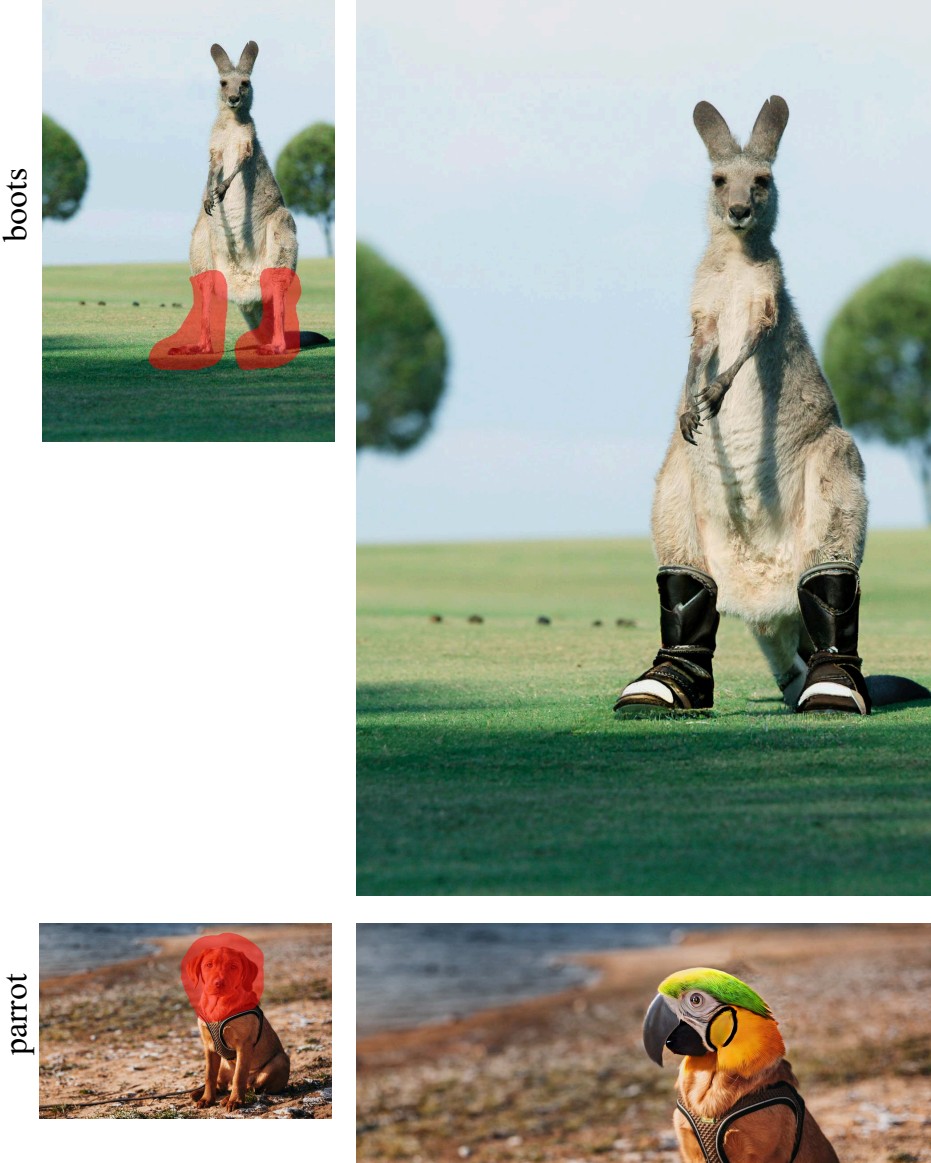

Figure 23: More high-resolution results of our method. Zoom in to view high-resolution details.

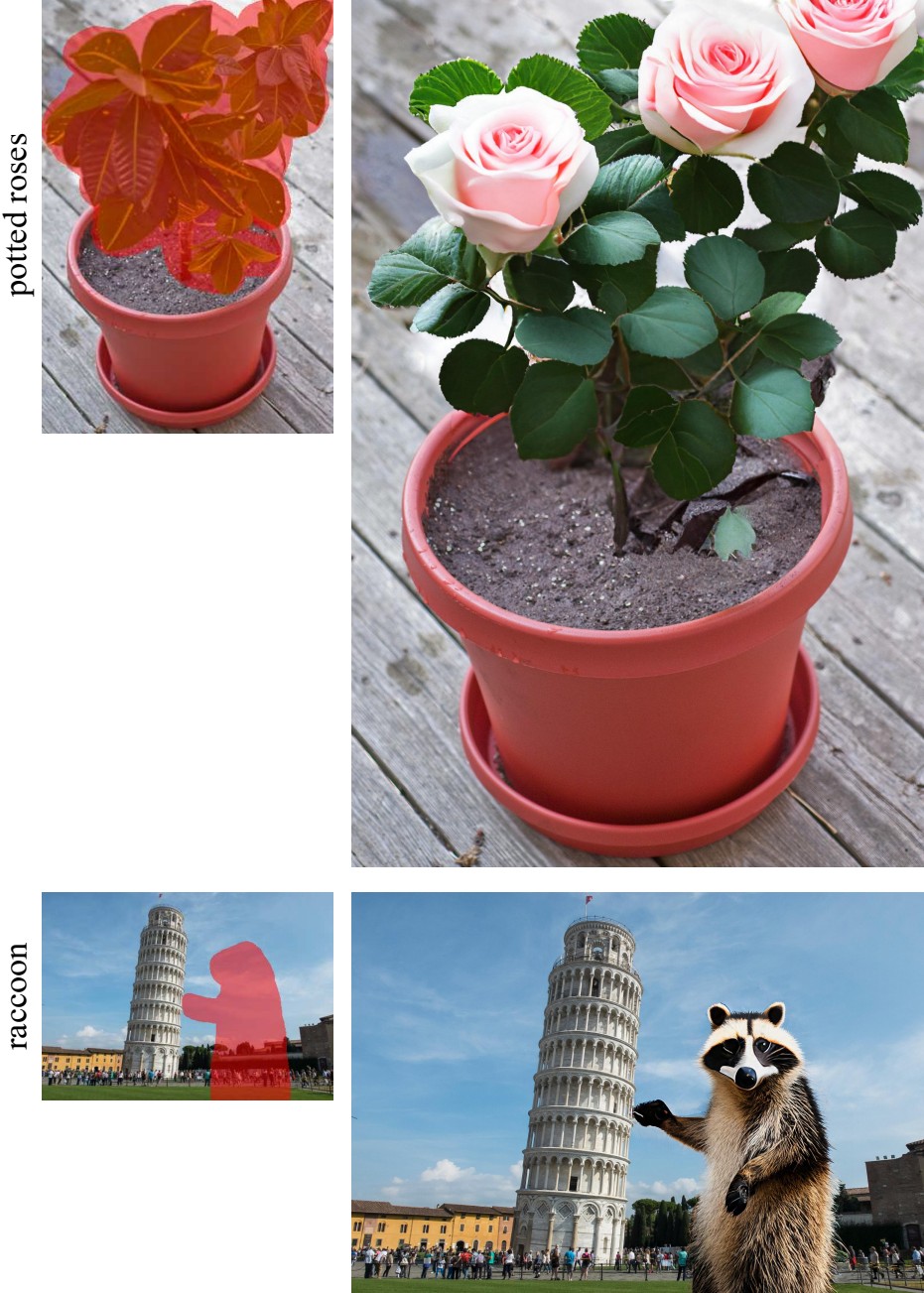

Figure 24: More high-resolution results of our method. Zoom in to view high-resolution details.

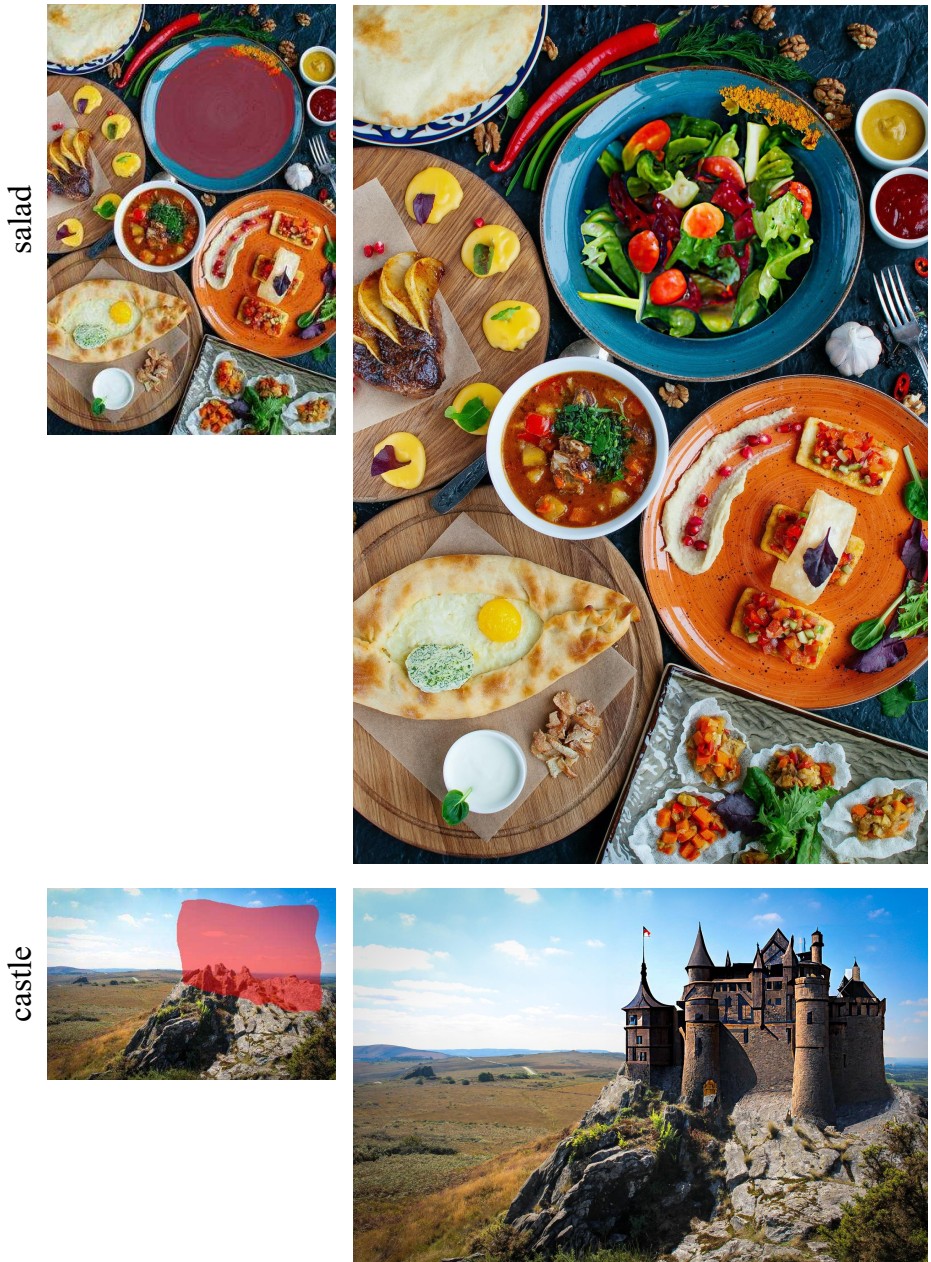

Figure 25: More high-resolution results of our method. Zoom in to view high-resolution details.

