# OpenReview forum: "HD-Painter: High-Resolution and Prompt-Faithful Text-Guided Image Inpainting with Diffusion Models"
_ICLR.cc/2025/Conference — ICLR 2025 Poster_

### Official Review · Reviewer_Te6W · 2024-10-23

**Soundness:** 3
**Presentation:** 3
**Contribution:** 3
**Rating:** 6
**Confidence:** 3

**Summary:**

The paper investigates the diffusion based image inpainting task. The proposed method is training-free, which modifies the self attention block and use post training alignment/guidance. The experimental results prove the effectiveness of the proposed method.

**Strengths:**

- The method is intuitively clear and reasonable. The self attention may contain irrelevant information regarding area to be inpainted. The modification to self attention block makes sense. The visualization of attention map proves the efficiency of the method.
- Although a bit heuristic, the training-free nature makes the method easily extensible to other pretrained inpainting diffusion models.
- The proposed RASG is cleverly simple yet effective, which transforms the post training alignment to the form of non-deterministic DDIM. It seems to efficiently avoid noisy latent deviating too far from the original trajectory.

**Weaknesses:**

- Lack of experiments on running time. What is the additional time cost associated with the proposed method?
- Lack of ablation study on hyperparameters. How sensitive is the model to different hyperparameters? Espscially $\eta$.

**Questions:**

Please check the weakness.

(Minor comment) In line 183, it should be VQGAN or VAE within SD, where usually VAE is chosen.

---

> ### Author Response · Authors · 2024-11-22
> **Response to Reviewer Te6W**
>
> **[Q1]** Thank you for the remark. We have added the runtime report to the Implementation Details Section in the main paper. It is as follows
>
> > In PAIntA’s implementation, we reuse calculated cross-attention similarity maps, which results in a very small performance impact. With PAIntA the model is about just $10 \\%$ slower, making $\sim 3.3$ seconds from $\sim 3$ seconds of the baseline.
> >
> > For RASG, naturally, the backward pass of the model increases the runtime about twice.
> However, optimizations, like using RASG only for a subset of steps, etc., can potentially greatly decrease the runtime while keeping the generation distribution. We keep such investigations for future research.
>
> **[Q2]** Thank you for the comment, we performed the ablation study on the $\eta$ hyperparameter and observed that RASG provides improvements for various values of $\eta$:
>
> | Model Name                 | CLIP score ↑ | Accuracy ↑   | Aesthetic score ↑ |
> | -------------------------- | ------------ | ------------ | --------------- |
> | DS8 (DreamShaper 8) | 25.61 ± 0.02 | 58.93 ± 0.18 | 4.965 ± 0.004 |
> | DS8+ProFI-Painter (eta=0.10) | 26.26 ± 0.03 | 67.44 ± 0.63 | 4.987 ± 0.005   |
> | DS8+ProFI-Painter (eta=0.15) | 26.32 ± 0.03 | 68.05 ± 0.48 | 4.980 ± 0.003   |
> | DS8+ProFI-Painter (eta=0.20) | 26.36 ± 0.05 | 68.08 ± 0.42 | 4.969 ± 0.004   |
>
> The table above shows that our choice $\eta=0.15$ demonstrates a good tradeoff between high accuracy, CLIP-score, and high aesthetic score.
>
> **(Minor comment)** Thank you for noticing. This has been fixed in the revision.
>
> We appreciate the reviewer's valuable comments and positive feedback and hope our response contributes to the comprehensiveness of our work.

---

> > ### Comment · Reviewer_Te6W · 2024-11-25
> >
> > Thank the authors for the rebuttal, according to the response, I keep my initial score.

---

### Official Review · Reviewer_Lgey · 2024-10-31

**Soundness:** 3
**Presentation:** 2
**Contribution:** 3
**Rating:** 6
**Confidence:** 3

**Summary:**

Paper addresses the problem of prompt neglect in text-guided image inpainting. Existing solutions (smartbrush, imagen editor) are argued to have reduced generation quality. The proposed method is training-free. The paper proposes to techniques, Prompt-aware Introverted Attention layer, and Reweighting Attention Score guidance for accurate prompt following and high image quality.

**Strengths:**

- the hypothesis that the problems of existing methods is based because of the self-attention, and that this problem can be addressed there, is interesting and convincing (even though it is not well explained).
- the qualitative results clearly show the superiority of the proposed method. Also the quantitative results are ok, but they do not seem to indicate the large improvement seen in the qualitative results.

**Weaknesses:**

---
- the self-attention map analysis (in Appendix B) is important for the motivation of the paper and should be moved to main paper. This would help to provide a motivation in section 3.3. before just stating it in math, helping the reader understand the proposed method.

- the explanation of the main idea behind section 3.3 is not well presented. The main idea is the introduction of c_j in the self attention, but what this represent is not well explained in words.

- does the 'introverted' nature make it hard to use information from outside of the impainted region. For example if you ask for an object with a whole behind which the background should continue ? Or a bike in front of a fence, etc.

- section 3.4 should also start out by stating the problem it addresses and how it is planning to address this. I found the presentation not good of this section, and very hard to understand.

- in the user study the results of DreamShaper are better than SmartBrush, but in figure 5 the results of DreamShaper are very bad not following the prompt at all, and SmartBrush is much better. Any explanation, this makes me doubt the correctness/usefullness of the user study.

MINOR points:

Think would be better to directly put equation of c_j also in (5), and then explain. Try to first explain the main idea, then the details (SOT, EOT, clipping etc). Now the main idea is hard to distill.

More usage of \citep might make reading easier (e.g. line 96).

Too many forward references in introduction (to future tables and figures and appendices).

References for relevant information to appendix are out of place in the introduction. The main information should be in the main paper; the introduction introduces the most relevant information of the paper.

line 248. Maybe better to keep professional factual style (instead of diary-style 'we did this' 'than that'), so better 'a thorough analysis of Stable Inpainting led to the conclusion...'

**Questions:**

Overall I found the visual results appealing. The quantitative improvement less so (maybe a new metric should be developed to better show the superiority of the method ?). I found the presentation of the crucial section 3.3-3.4 of bad quality and they need to be improved much.

- see weaknesses.

---

> ### Author Response · Authors · 2024-11-22
> **Response to Reviewer Lgey**
>
> **[Q1]** Thank you for the suggestion. We moved the self-attention map analysis from Appendix B to the beginning of Section 3.3. We then discuss how the vanilla self-attention heatmaps show high similarity between generated and background pixels, proving that the model over-concentrates on creating regions visually similar to the existing ones, while disregarding the prompt. Finally, we introduce the intuition behind PAIntA, and discuss why it solves the mentioned issue. Only then we proceed to a deep-dive.
>
> **[Q2]** We added more explanation on $c_j$ in Section 3.3. In particular, before the formal mathematical definition we added the following:
>
> > $c_j$ represents the amount of how much we want to suppress the impact of the known region pixel $j$ on the completion of the missing region. As we want the generation in the missing region to be more aligned with the provided textual prompt, we set $c_j$ based on the similarity of $j$ and the prompt in the embedding space. In other words, we set $c_j$ low for such pixels $j$ from the known region that are not semantically close to the given prompt, and we set $c_j$ high otherwise.
>
> We hope this explanation makes the main idea more intuitive and easier to understand.
>
> **[Q3]** PAIntA is designed to reduce the amount of information from the outside of the inpainting region, but does not completely remove it. However, we performed a small experiment to verify that our method can handle holes inside of the masked area. You can find the results in this [anonymous link](https://anonymous.4open.science/r/a32a709b6fdf-1D67/R3Q3%20Introverted%20Holes.png).
>
> As can be seen, the ability to use the outside context remains when using PAIntA, e.g. black dot pattern is preserved from the background to the hole of the donut in the second example, and the fence is continued from the background to the generated region in the fourth example with a bicycle.
>
> **[Q4]** Thank you for the feedback and the suggestion. We improved the writing by making several changes to this section. We now start by highlighting the issue, and suggesting to solve it with post-hoc guidance. We then discuss the issues of vanilla post-hoc guidance, and how RASG helps to alleviate them, and only then introduce the mathematical details.
> We also added a small discussion of our intuition for choosing the guidance objective $S(x)$ before jumping to the definitions.
>
> **[Q5]** We have double checked with the user study participants, and figured out that they were selecting the best results based on several criteria. Particularly in the case of prompt alignment if the generated object does not have one of the attributes described in the prompt (such as the duck is not white in the SmartBrush generation in Fig. 5) they don’t mention it as the best. In Fig. 5 SmartBrush have 3 such generations: the duck is not white, the vase is less “ancient greek” than of ProFI-Painter (due to the patterns specific to such vases, according to one of the participants we double-checked with), and the boat is less similar to a boat than in the case of ProFI-Painter. And also, if all attributes are correctly generated but just with small sizes, they were treating the generations as good in terms of prompt alignment. Such examples in Fig. 5 are the small sunglasses of the owl and the small flames of the car generated by DreamShaper.
>
> Additionally, the user study is conducted in a way that for each example the participants choose the best result as the winner, so for the cases when ProFI-Painter is the best, all the rest are treated as equivalently worse. Therefore, the user study is majorly informative when revealing the best method among all, and less informative when comparing two non-best approaches.
>
> **Minor points**
>
> **[Minor Q1]** After revising the writing, we now first discuss the idea behind the construction of the factors $c_j$, namely that $c_j$ should represent the amount of how much we want to suppress the impact of a known region pixel j on the generation process of the pixels in the unknown region, and only then give the formal definition of $c_j$. We hope this change helps with the clarity of the text.
>
> **[Minor Q2]** Thank you for the suggestion. We went over the whole text and made changes including the usage of \citep where appropriate.
>
> **[Minor Q3]** We removed redundant references and kept only those essential for understanding the main issues in existing methods that ProFI-Painter aims to address.
>
> **[Minor Q4]** We thank the reviewer for noticing this, and we moved the corresponding part to Sec. 3, so the introduction no longer contains references to Appendix.
>
> **[Minor Q5]** We thank the reviewer for this comment and following the suggestion we improved the writing style.
>
> We thank you for comments and suggestions. We hope our explanations clarify the questions above.

---

> ### Author Response · Authors · 2024-11-27
>
> Dear Reviewer Lgey,
>
> Thank you for your detailed review and invaluable suggestions. We highly appreciate the effort and time you have dedicated to reviewing our paper. Based on your comments, we have revised the paper by incorporating the necessary changes and updated the PDF. We also carefully addressed your concerns in our previous response.
>
> With the discussion period deadline approaching, we would be grateful if you can review our responses and share any further questions or concerns you might have.
>
> Thank you once again.
>
> Best regards,
>
> Authors.

---

> > ### Comment · Reviewer_Lgey · 2024-11-29
> >
> > Dear authors, sorry for late reply. Thank you for your much improved revision and clarifications. My concerns have been addressed, I think, motivation and presentation of sections in paper has much improved. I therefore am happy to raise the score to a 6 a support acceptance.

---

### Official Review · Reviewer_Bisp · 2024-11-01

**Soundness:** 3
**Presentation:** 2
**Contribution:** 3
**Rating:** 6
**Confidence:** 4

**Summary:**

This paper introduces a training-free approach to enhancing prompt-guided image inpainting with diffusion models. It proposes two key components: Prompt-Aware Introverted Attention (PAIntA) and Reweighting Attention Score Guidance (RASG), which improve alignment with text prompts. PAIntA adjusts self-attention layers to prioritize text-related regions, while RASG refines cross-attention scores for better prompt consistency. A specialized super-resolution technique ensures high-quality image scaling. Quantitative and qualitative results on MSCOCO confirm the method’s superiority.

**Strengths:**

The discussion about prompt neglect is promising.

The proposed solution achieves strong results on evaluation metrics.

**Weaknesses:**

Some discussion and analysis should be included, see the question part.

**Questions:**

The inference time cost should be reported and compared.

How to derive Claim 1? How to define high-quality images?

Will the proposed method work on transformer-based models like SD3 and FLUX?

In Table 2, the bolded aesthetic score is not the best one.

---

> ### Author Response · Authors · 2024-11-22
> **Response to Reviewer Bisp**
>
> **[Q1]** Thank you for this remark, we added a report on the inference time in the revised paper (Sec. 4.1). In particular, we added the following.
>
> > In PAIntA’s implementation, we reuse calculated cross-attention similarity maps, which results in a very small performance impact. With PAIntA the model is about just $10 \\%$ slower, making $\sim 3.3$ seconds from $\sim 3$ seconds of the baseline.
> >
> > For RASG, naturally, the backward pass of the model increases the runtime about twice.
> However, optimizations, like using RASG only for a subset of steps, etc., can potentially greatly decrease the runtime while keeping the generation distribution. We keep such investigations for future research.
>
> **[Q2]** The derivation of Claim 1 from the Theorem 1 of Song et al. [1] can be found in the same paper [1], beginning of Sec. 4 (intro and Sec. 4.1). Our Eq. (3) just repeats Eq. (12) of [1].
>
> In short, Theorem 1 of Song et al. claims that in order to minimize $J_\sigma$ objective, allowing to sample according to their Eq. (12)
> $$
> x_{t-1} = \sqrt{\alpha_{t-1}} \frac{x_t - \sqrt{1 - \alpha_t}\epsilon_\theta^{t}(x_t)}{\sqrt{\alpha_t}}  + \sqrt{1 - \alpha_{t-1} - \sigma_t^2} \epsilon_\theta^t(x_t) + \sigma_t \epsilon_t,
> $$
> it is sufficient to minimize the DDPM objective $L_\gamma$. So for already pretrained DDPM models the sampling Eq. (12) will make sense.
>
> By “... can be applied to generate high-quality images” we mean that sampling with Eq. (12) makes sense (hence will give plausible results) as $J_\sigma$ is minimized.
>
> **[Q3]** Thank you for the question. Since FLUX is based on SD3 paper and is a later and better model, we opted to try our method for FLUX and have observed a positive impact both qualitatively and quantitatively. In particular, the generation accuracy improved from $58.32 \\%$ to $65.31 \\%$ (on the same test set of 10K MSCOCO images we used in the paper) after adding ProFI-Painter to a FLUX-based inpainting, showing a significant boost in prompt alignment. In addition, we performed a qualitative analysis on the same visual  test set from our paper and validated the generation improvement, some visual comparison can be found in this [anonymous link](https://anonymous.4open.science/r/a32a709b6fdf-1D67/R1Q1%20FLUX%20Inpainting.png).
>
> Also, since Reviewer sKdN was asking the same question also about SDXL, we added our method on top of SDXL-inpainting as well. Similar to the FLUX case, here we also observed a positive impact: the generation accuracy improved from  $52.98  \\%$ to $63.58  \\%$, and, qualitatively, the visual comparison, presented in this [anonymous link](https://anonymous.4open.science/r/a32a709b6fdf-1D67/R1Q1%20SDXL%20Inpainting.png), validates the improvement in prompt-alignment.
>
> In addition, in the response to the first question of Reviewer sKdN, we describe how we adapted ProFI-Painter’s components, PAintA and RASG, for FLUX, as this process may not seem straightforward. In order to not repeat the same response here, we kindly refer the reviewer to our response above ([Q1] of Reviewer sKdN) for more details.
>
> **[Q4]** Thank you for noticing. It is now fixed in the revision.
>
> We thank the reviewer for the valuable feedback and the positive rating. We hope our response clarifies the questions remained.
>
> **References**
>
> [1] J. Song, C. Meng, S. Ermon, “Denoising Diffusion Implicit Models”, in ICLR 2021

---

> > ### Comment · Reviewer_Bisp · 2024-11-25
> >
> > Thank you for your response. Most of my concerns are addressed.

---

### Official Review · Reviewer_sKdN · 2024-11-01

**Soundness:** 3
**Presentation:** 3
**Contribution:** 3
**Rating:** 6
**Confidence:** 3

**Summary:**

The authors introduced Prompt-Aware Introverted Attention (PAIntA) block without any training or fine-tuning requirements, enhancing the self-attention scores according to the given textual condition aiming to decrease the impact of non-prompt-relevant information. They also proposed Reweighting Attention Score Guidance (RASG), a post-hoc mechanism seamlessly integrating the gradient component in the general form of DDIM process. This allows to simultaneously guide the sampling towards more prompt-aligned latents and keep them in their trained domain.

**Strengths:**

* The writing is clear to understand with detailed formulations and figures.

* The results are superior when compared to other methods.

* The phenomena of Appendix B are very interesting, revealing that the original model maintains a similar visual pattern from other parts of images and the PAIntA would increase the probability to respond to the prompts.

**Weaknesses:**

Some questions below

* Would this method be easily adapted to some modern models, for example, SDXL, SD3, or even FLUX?

* The successful rate of one case with sufficient sampling of different seeds is not clear.

* The experiments are conducted in cases with few instances, so if there are multiple instances (>5), what is the performance?

* Could the method deal with the inpainting tasks with multiple masks in one inference?

**Questions:**

Please see weaknesses.

---

> ### Author Response · Authors · 2024-11-22
> **Response to Reviewer sKdN [Q1]**
>
> **[Q1]** Thank you for the question. Our method, ProFI-Painter, introduces 2 contributions: PAIntA and RASG. Requirements for PAIntA integration is the existence of self-attention layers in the predictor network and a means to measure the similarity of a given spatial location to the textual prompt, while the requirement for RASG application is the possibility of sampling with a DDIM sampler. SDXL, SD3 and FLUX meet those requirements, therefore the application of our approach is possible on inpainting methods based on those text-to-image models. Since FLUX is based on SD3 paper and is a later and better model, we opted to conduct our ProFI-Painter experiments on SDXL and FLUX-based text-guided image inpainting models.
>
> Quantitatively, we measured the generation accuracy on our test set of 10K images (from MSCOCO dataset) in order to see the impact. The generation accuracy of SDXL-inpainting vs SDXL-inpainting + ProFI-Painter was $52.98  \\%$ vs $63.58  \\%$, and for FLUX-inpainting vs FLUX-inpainting + ProFI-Painter was $58.32 \\%$ vs $65.31 \\%$. Both show significant boosts in generation accuracy when used in combination with ProFI-Painter demonstrating the effectiveness and the universality of our approach.
>
> Qualitatively, we compared both settings on our visual test set from the paper and in this [anonymous link](https://anonymous.4open.science/r/a32a709b6fdf-1D67/R1Q1%20SDXL%20Inpainting.png) we share several examples for SDXL-inpainting vs SDXL-inpainting + ProFI-Painter, and in [this link](https://anonymous.4open.science/r/a32a709b6fdf-1D67/R1Q1%20FLUX%20Inpainting.png) for FLUX-inpainting vs FLUX-inpainting + ProFI-Painter. It can be clearly seen that  ProFI-Painter helps both methods to generate prompt-aligned results with high quality.
>
> Additionally, below we describe how we adapted ProFI-Painter’s components, PAintA and RASG, for FLUX, as this process may not seem straightforward. In the case of SDXL, applying PAintA and RASG is straightforward as SDXL is also based on a similar UNet architecture as Stable Diffusion 1.5 and 2, for which our method is thoroughly described in the paper.

---

> > ### Author Response · Authors · 2024-11-22
> > **Details on FLUX-based ProFI-Painter**
> >
> > **FLUX-based ProFI-Painter:**
> >
> > As the authors of FLUX haven't released an official FLUX-inpainting method at the moment of making this report, we first needed to fine-tune the official [FLUX.1-dev](https://huggingface.co/black-forest-labs/FLUX.1-dev) text-to-image model on the inpainting task before proceeding to ProFI-Painter experiments on that baseline. To that end, we modified the FLUX architecture similar to the modification of Stable Inpainting over Stable Diffusion and trained a new FLUX-inpainting model.
> >
> > Then to make ProFI-Painter work with FLUX-inpainting model we applied PAIntA and RASG as follows.
> >
> > **PAIntA:**
> >
> > We incorporate PAIntA into FLUX’s self-attention layers which operate on the concatenated sequence of textual and image features. Let $X^{img}\in \mathbb{R}^{H\times W\times C}$ and $X^{txt}\in\mathbb{R}^{L\times C}$ be these feature groups respectively. As in the case of Stable Inpainting, here also PAIntA considers such $X^{img}_j$ features (pixels) that are from the known region and suppresses their impact (attention scores) on the generated region image features (pixels) $X^{img}_i$. The suppression is done by multiplicating the attention scores between $X^{img}_j$ and $X^{img}_i$ by a coefficient $c_j\in [0,1]$. The suppression coefficient $c_j$ for the known region feature $X^{img}_j$ is being chosen based on its similarity to the prompt which we compute by averaging the attention scores of $X^{img}_j$ with the textual features $X^{txt}$. To keep $c_j$ in the interval [0,1] we later normalize and clip exactly as done in the case of Stable Inpainting + PAIntA (discussed in Sec. 3.3 of our paper).
> >
> > **RASG:**
> >
> > To apply RASG post-hoc guidance strategy for FLUX we adapt the optimal transport sampling of FLUX’s flow-matching approach to the DDIM sampling approach. That is we derived the noise prediction $\epsilon^t_{\theta}(\cdot)$ based on FLUX’s velocity prediction $v_{\theta}(\cdot, t)$ and used the RASG equation (8) from our (revised) paper (for some objective function S(x)):
> > $$
> > x_{t-1} = \sqrt{\alpha_{t-1}} \frac{x_t - \sqrt{1 - \alpha_t}\epsilon^t_\theta(x_t)}{\sqrt{\alpha_t}} +
> > \sqrt{1 - \alpha_{t-1} - \sigma_t ^ 2} \epsilon^t_\theta(x_t) +
> > \sigma_t \frac{\nabla_{x_t}S(x_t)}{\mbox{std}(\nabla_{x_t}S(x_t))}. \quad\quad\quad (8)
> > $$
> > We know that FLUX’s model tries to predict the velocity field $v_{\theta}((1-t)x_0 + t\varepsilon) \approx \varepsilon - x_0$, where $x_0$ is a sample from data, and $\varepsilon\sim \mathcal{N}(0,I)$.
> >
> > Note that here the perturbation is linear: $x_t = (1-t)x_0 + t\varepsilon$, while the DDIM sampling with RASG guidance mentioned above is designed for variance-preserving diffusion processes. Therefore, if we define another diffusion process
> > $$
> > x_t^{\prime} = \frac{1-t}{\sqrt{(1-t)^2+t^2}}x_0 + \frac{t}{\sqrt{(1-t)^2+t^2}}\varepsilon = \frac{x_t}{\sqrt{(1-t)^2+t^2}},
> > $$
> > the latter will be a variance-preserving perturbation (as in the case of DDPM / DDIM) with $\alpha_t = \frac{(1-t)^2}{(1-t)^2+t^2}$. Additionally, since $\epsilon^t_{\theta}(x_t^{\prime})$ approximates the noise $\varepsilon$, and $\varepsilon = x_t + (1-t)(\varepsilon - x_0) \approx x_t + (1-t)v_{\theta}(x_t, t)$, we get the following relation between $\epsilon^t_{\theta}(\cdot)$ and $v_{\theta}(\cdot, t)$:
> > $$
> > \epsilon^t_{\theta}\left(\frac{x_t}{\sqrt{(1-t)^2 + t^2}}\right) = x_t + (1- t) v_{\theta}(x_t, t).
> > $$
> > For using RASG, it remains to use $\alpha_t, \epsilon^t_{\theta}(\frac{x_t}{\sqrt{(1-t)^2 + t^2}})$  and we will get the sample for $x_{t-1}^{\prime}$. Finally, for obtaining $x_{t-1}$, we rescale the variance-preserving diffusion latent $x_{t-1}^{\prime}$ and get $x_{t-1} = x_{t-1}^{\prime} \sqrt{(1-t)^2 + t^2}$.
> >
> > $\sigma_t$ are chosen with Eq. (11) from the (revised) paper, and the post-hoc objective function $S(x)$ is chosen in the same way as in the paper  (see Sec. 3.4).

---

> ### Author Response · Authors · 2024-11-22
> **Response to Reviewer sKdN [Q2-Q4]**
>
> **[Q2]** We conducted an additional experiment to evaluate the success rate for a larger number of seeds. We used 15 different seeds to inpaint all 10K MSCOCO examples in our test set. To determine whether a generated example is successful, we employ an object detection model on the output image, limited to the bounding box of the mask, and see if the object in the prompt exists in the list of detected objects.
> The success rate is the number of images marked as successful divided by 15. This way, the average success rate for all 10K images in the set is $87.01\\%$.
>
> For further clarity, you can refer to this [anonymous link](https://anonymous.4open.science/r/a32a709b6fdf-1D67/R1Q2%20Many%20Seeds.png) for visual examples. Here we show all 15 outputs for a selected subset of the images. As you can see, across the examples, the average success rate is approximately the number reported above.
>
>
> **[Q3]** We assume the question refers to having multiple instances of the target object present in the input image. We examined the behavior of our model in this case, and added the corresponding results to our appendix. Please refer to Appendix F, or use [this link](https://anonymous.4open.science/r/a32a709b6fdf-1D67/R1Q3%20Mulitple%20Instances.png). As can be noticed from the examples, our approach successfully handles the cases when there are multiple instances of the same object and is able to generate a new one.
>
>
> **[Q4]** We added another appendix section (Appendix G), with visual examples (also [here](https://anonymous.4open.science/r/a32a709b6fdf-1D67/R1Q4%20Multiple%20Objects.png)) of cases with multiple masks. We analyzed such cases and came to the conclusion that our approach is able to generate visually appealing results for multi-component masks.
>
> We would like to thank the reviewer for the feedback and the positive rating, and hope that our response will help to further clarify the questions that remained.

---

> > ### Comment · Reviewer_sKdN · 2024-11-23
> >
> > Thank you for your detailed response. My questions have been addressed. However, as I am not an expert in this field, I find it challenging to provide a more advanced review. Therefore, I will maintain my current score.

---

### Meta-Review · Area_Chair_NA9K · 2024-12-22

**Metareview:**

This paper presents ProFI-Painter, a training-free approach that accurately follows prompts for text-to-image inpainting. Moreover, they also proposed RASG, a post-hoc mechanism to guide the sampling towards latents aligned to prompts and prevent out-of-distribution latent shifts.

The major paper strengths: 1) The discussion on prompt neglect for inpainting and the proposed idea to improve prompt following is interesting.  2) Both ProFI-Painter and RASG are effective. 3) The experiments show promising results.  4) The extension to FLUX in authors' rebuttal is also helpful.

Considering these strengths, I will recommend "Accept (poster)".

**Additional Comments On Reviewer Discussion:**

Multiple reviewers asked a out whether the proposed method can be extended to recent T2I methods like FLUX, and authors extended their approach to FLUX in the rebuttal.

In their rebuttal, the authors also addressed the questions on experiment part (e.g., multi instance, multi masks, ablation study, runtime analysis), or clarification on some technical parts.

---

> ### Public Comment · ~Shant_Navasardyan1 · 2025-02-15
> **Request to Modify the Paper Title for the Camera-Ready Version**
>
> Dear Program Chairs, Area Chair, and reviewers.
>
> Thank you for your time and effort in reviewing our paper and providing valuable feedback.
>
> We would like to make a minor change in the title of our paper in the camera-ready version to align more with our open-source efforts. In particular, we want to change the name of the method from ProFI-Painter to HD-Painter, by so changing the title from "ProFI-Painter: Text-Guided Prompt-Faithful Image Inpainting with Diffusion Models" to "HD-Painter: High-Resolution and Prompt-Faithful Text-Guided Image Inpainting with Diffusion Models".
>
> Please let us know if you have any objections or suggestions regarding this change.
>
> Best regards,
> Authors.

---

### Decision · Program_Chairs · 2025-01-22

Accept (Poster)